

# Quantification of CO emissions from the city of Madrid using MOPITT satellite retrievals and WRF simulations

Iris Dekker[1,2], Sander Houweling[1,2], Ilse Aben[1], Thomas Röckmann[2], Maarten Krol[1,2,3], Sara Martínez-Alonso[4], Merritt Deeter[4], and Helen Worden[4]

[1]SRON Netherlands Institute for Space Research, Utrecht, The Netherlands
[2]Institute for Marine and Atmospheric Research Utrecht, Utrecht University, The Netherlands
[3]Department of Meteorology and Air Quality, Wageningen University and Research Centre, Wageningen, The Netherlands
[4]National Center for Atmospheric Research (NCAR), Boulder, CO, USA

*Correspondence to:* Iris Dekker (i.n.dekker@uu.nl)

**Abstract.** The growth of mega-cities leads to air quality problems directly affecting the citizens. Satellite measurements are becoming of higher quality and quantity, which leads to more accurate satellite retrievals of the enhanced air pollutant concentrations over large cities. In this paper, we compare and discuss both an existing and a new method for estimating urban scale trends in CO emissions using multi-year retrievals from the MOPITT satellite instrument. The first method is mainly based on

satellite data, which has the advantage of fewer assumptions, but also comes with uncertainties and limitations as shown in this paper. To improve the reliability of urban to regional scale emission trend estimation, we simulate MOPITT retrievals using the Weather Research and Forecast model with chemistry core (WRF-Chem). The difference between model and retrieval is used to optimize CO emissions in WRF-Chem, focusing on the city of Madrid, Spain. This method has the advantage over the existing method in that it allows both a trend analysis of CO concentrations and a quantification of CO emissions. Our analysis

confirms that MOPITT is capable of detecting CO enhancements over Madrid, although significant differences remain between the yearly averaged model output and satellite measurements ($R^2$=0.75) over the city. After optimization, we find Madrid CO emissions to be lower by 48% for 2002 and by 17% for 2006 compared with the EdgarV4.2 emission inventory. The MOPITT derived emission adjustments lead to better agreement with the European emission inventory TNO-MAC-III for both years. This suggests that the downward trend in CO emissions over Madrid is overestimated in EdgarV4.2 and more realistically rep-

resented in TNO-MAC-III. However, our satellite and model based emission estimates have large uncertainties, around 20% for 2002 and 50% for 2006.

## 1 Introduction

During the last decades, global urbanisation has led to an increase in the number of large cities. Several hundred cities currently have more than a million inhabitants. These highly populated cities with dense traffic networks are important sources of many

kinds of air pollutants that directly affect the large fraction of the population living there (e.g., Pascal et al. (2013); Kan et al. (2012); Romero-Lankao et al. (2013)). Therefore, global urbanisation increases the need for air quality monitoring and prediction in large cities. Large cities are also important sources of several greenhouse gases (GHGs). A recent development



in air quality and GHG monitoring is the use of sensors on board of satellites to augment ground-based measurement networks in cities. Especially in cities without a dense measurement network, satellite data can have an important added value. Thanks to improvements in the quality and spatial resolution and sampling of data from atmospheric conposition sensors on board of satellites over the past decades, detection and quantification of city emissions is becoming feasible for an increasing number

of air pollution species (Streets et al., 2013). Nitrogen oxides (NO and $NO_2$, together called $NO_x$) emissions from cities have been successfully quantified in several studies (e.g., Beirle et al. (2011); Liu et al. (2016)). First steps have also been made to quantify urban emissions of other species such as sulphur dioxide using satellite observations (Fioletov et al., 2011). Urban carbon monoxide (CO) has also been studied (Pommier et al., 2013; Clerbaux et al., 2008; Worden et al., 2012); this paper is focused on quantifying urban CO emissions.

CO is a major air pollutant in cities. It is a toxic gas for humans at ground level, although at mole fractions that are substantially higher than those found under normal conditions (usually < 1ppm in most present-day urban environments). The World Health Organisation recommends a maximum of 9 ppm CO for eight hour exposure (WHO, 2004). CO is an important precursor of tropospheric ozone and a primary control on the oxidizing power of the atmosphere. The primary sink of CO is the hydroxyl radical (OH). The lifetime of CO varies between several weeks and several months, depending on the location

and season (e.g., Holloway et al. (2007); Khalil and Rasmussen (1990)). The relatively long lifetime compared to some other air pollutants results in a rather smooth spatial distribution. Therefore the difference in concentration between the background atmosphere and regions close to sources is smaller than for $NO_x$, which has a lifetime of hours to days. This makes CO sources harder to detect and quantify than $NO_x$ sources. On the other hand, urban CO is easier to detect than urban carbon dioxide and methane, for example, which have lifetimes of several years to decades leading to well-mixed distributions and relatively

small source signals. Due to its intermediate lifetime, CO can be a good tracer of pollution transport and has been used, for example, as a proxy for anthropogenic emissions of the important greenhouse gas carbon dioxide (Gamnitzer et al., 2006). The increasing availability of CO measurements from Earth orbiting satellites raises the interest in the use of remote sensing for studying urban CO emissions.

The Measurement Of Pollution In The Troposphere (MOPITT) remote sensing instrument, on board the NASA Terra satel-

lite, has been measuring global CO concentrations since March 2000. The added value of MOPITT compared to other satellite instruments is that it can retrieve CO not only in the thermal infrared (~4.7$\mu$m) but also in the near infrared (~2.3$\mu$m) wavelengths, which together provide improved sensitivity to CO near the Earth's surface (Worden et al., 2010; Deeter et al., 2009). ESA's TROPOspheric Monitoring Instrument (TROPOMI) instrument, to be launched in 2017 on the Sentinel 5 precursor satellite, will also measure CO concentrations in this shorter wavelength range around 2.3$\mu$m (Landgraf et al., 2016; Fu et al.,

2016; Abida et al., 2017). The TROPOMI spatial resolution, 7x7 $km^2$ at nadir and daily global coverage, will be increased considerably compared to MOPITT which has 22x22 $km^2$ spatial resolution and global coverage once every 2-3 days (Edwards et al., 2004), making it even more suitable for city emission estimation.

So far, satellite retrievals of CO have been used mainly in global scale analyses, quantifying large-scale CO emissions (e.g., Hooghiemstra et al. (2012a); Leeuwen van et al. (2013); Hooghiemstra et al. (2012b); Girach and Nair (2014); Yin et al. (2015);

Jiang et al. (2017)) with a primary interest in biomass burning. Furthermore, the first attempts have been made to use MOPITT



CO retrievals to quantify emissions from cities (Pommier et al., 2013; Clerbaux et al., 2008). These studies demonstrated that CO pollution plumes over large cities can be distinguished from the background in satellite data. However, averaging over long time periods was necessary to reduce measurement noise. In addition, Pommier et al. (2013) calculated relative trends in CO emissions from changes in the observed CO enhancement over cities in time. However, to move from this estimation of relative

trends to the quantification of the emissions requires additional information on atmospheric dispersion.

The aim of this work is to estimate CO emissions from cities by quantifying the relationship between local concentration enhancements and emissions, making use of the Weather Research and Forecasting (WRF) model together with the MOPITT retrievals. The method is developed in a way that can easily be applied to other satellite data as TROPOMI data; we expect the robustness of the method to increase when used with the higher sampling and finer spatial resolution of the TROPOMI data. We

test the performance of this method in comparison with the method for estimating emission trends using only satellite data of Pommier et al. (2013), which we will refer to as the "satellite-only" method, focusing on specific aspects that can influence the estimation of emission trends using the satellite-only method that do not influence the emission estimation in our own method.

For the satellite-only method, we investigated nine target cities: Baghdad, Delhi, Los Angeles, Mexico City, Moscow, Paris, Sao Paulo, Tehran and Madrid. For our new method, referred to as "WRF optimization", we focus on the city of Madrid.

Madrid is a source for which two high-resolution emission inventories are available (Kuenen et al., 2014; Crippa et al., 2016) and which, due to its climate and isolated position from other sources has relatively favourable conditions for the retrieval of CO using MOPITT; this makes Madrid well suited for developing and testing the new method.

This paper is organized as follows: section 2 describes the MOPITT data and two methods to estimate emission (trends): the satellite-only method and our own WRF optimization technique. It also includes a brief description of the WRF model.

We then present results based on the satellite-only technique (section 3.1), and analyse its limitations (section 3.2). Next we describe the results of the WRF optimization method (section 3.3) and the analysis of its limitations (section 3.4). Finally, an outlook is presented in section 4, followed by the summary and conclusions in section 5.

## 2  Data and methods

### 2.1  MOPITT CO retrieval

MOPITT, on board the NASA Terra satellite, has been operating almost continuously since it was launched December 1999 in a sun synchronous orbit with a local equator crossing time of approximately 10.30 am / pm (Edwards et al., 2004). Data is available from March 2000 onwards. The size of pixels is 22 km x 22 km at nadir. The MOPITT swath is formed by scanning a four-pixel linear detector array across the satellite track and covers a total width of approximately 640 km. Neglecting the effects of clouds, near-global coverage takes about 3 days (Edwards et al., 2004). The MOPITT instrument uses gas correlation

radiometry to determine CO concentrations (Deeter et al., 2003). It has several instrument channels that sense infrared radiation (IR). The original MOPITT thermal infrared (TIR, ~4.7$\mu$m) retrieved CO dataset, has been expanded with Near Infrared (NIR,~2.3$\mu$m) retrievals (Deeter et al., 2009) and a combined NIR and TIR (hereafter called multispectral) product has been derived, with improved sensitivity to CO near the Earth's surface (Worden et al., 2010; Deeter et al., 2009). The multispectral



product combines the best features of both retrievals: higher sensitivity in the lower troposphere over land from the NIR, and vertical information in the free troposphere from the TIR (Deeter et al., 2014, 2013). The NIR channel adds most information in the lower troposphere and over land scenes with low thermal contrast (e.g. moist vegetation, (Deeter et al., 2009)). As the goal of our method requires maximum sensitivity to CO in the lower troposphere, we will mostly use the combined multispectral

CO retrievals.

For this research, MOPITT version 6 (and for comparisons with Pommier et al., 2013, version 5) level 2 data were used for the period March 2000 - December 2008 (Deeter, 2013a). The data of version 5 have been validated extensively (e.g., Deeter et al. (2013); Laat de et al. (2014)), and version 6 data have been validated by Deeter et al. 2014; 2016. The validation results showed that the version 6 data have reduced retrieval bias in the upper troposphere and confirms that the joint NIR and

TIR product has enhanced sensitivity to CO in the lower troposphere compared to the TIR only product. However, a negative concentration bias over the Amazon basin was reported in the version 6 multispectral product (Deeter et al., 2016). In version 6, compared to the previous version 5 data, a geolocation bias has been corrected (Deeter et al., 2014), and meteorological fields are derived from NASA MERRA instead of NCEP (Deeter et al., 2014). Monthly varying a priori data in version 6 are based on the CAM-CHEM model climatology for 2000-2009 gridded on $1°x1°$ (Deeter, 2013a), instead of the coarser gridded

MOZART climatology used in V5 and V4.

When using version 5 of the data, we corrected for the location bias in longitude using the formula also applied by Pommier et al. (2013, see Eq. 1). This method might give slightly different corrections from the corrections the MOPITT team applied to version 6 of the data (Deeter, 2012), especially in the temperate zones.

$$lon_{new} = lon_{orig} + 0.33 \times \cos(lat_{orig}) \tag{1}$$

In Eq. 1 $lon_{new}$ is the corrected longitude in radians, derived from the original coordinates ($lon_{orig}$, $lat_{orig}$; in radians). The NIR, TIR and combined multispectral data sets are made available on 10 pressure levels (surface to 100 hPa in 100 hPa intervals). The NIR product is not available for observations over oceans, or during night time overpasses (i.e., when the solar zenith angle exceeds 80 degrees). In this study these data are filtered out. Generally, the NIR product compared to the TIR product has relatively large random errors, requiring significant spatial and/or temporal averaging (Deeter, 2013b). The MOPITT retrieval,

especially the TIR part, has a varying vertical sensitivity. The monthly varying a priori CO climatology constrains the retrieved profile. The relative weights of the true atmospheric profile and a priori profile are represented by the Averaging Kernel (**AK**) matrix, which is made available for every retrieval. The relationship between the retrieved volume mixing ratio (VMR) profile ($\mathbf{x}_{rtr}$), true VMR profile ($\mathbf{x}_{true}$), a priori profile ($\mathbf{x}_a$) and averaging kernel matrix (**AK**) is given in Eq. 2.

$$\log_{10}(\boldsymbol{x}_{retr}) = \log_{10}(\boldsymbol{x}_a) + \boldsymbol{AK}(\log_{10}(\boldsymbol{x}_{true}) - \log_{10}(\boldsymbol{x}_a)) \tag{2}$$

the equation is logarithmic as the MOPITT retrieval algorithm assumes log-normal statistics for CO variability (Deeter, 2013a). Only daytime (solar zenith angle < 80°) and land pixels were used in this study to avoid a strongly varying influence of the NIR channel in the multispectral retrieval. In addition, retrievals were filtered for clouds, keeping data with a cloud description diagnostic value of 1 or 2. The cloud description diagnostic value is based on combined signals from MOPITT and MODIS




(Moderate Resolution Imaging Spectrometer, also on board of Terra) on cloud coverage, with a value of 1 indicating clear sky conditions according to MOPITT without information from MODIS, and a value of 2 indicating cloud free according to MOPITT and MODIS.

Due to the large pixel size of the MOPITT data relative to the size of cities, the long revisit time of the satellite, and the filtering on cloud free and daytime scenes, the number of useful data over individual cities was limited. Because the path of the urban pollution plume and background concentration field both vary strongly with meteorological conditions, it was necessary to average the MOPITT data temporally and spatially over a substantial time period to distinguish an urban signal from the background. The averaging technique of Fioletov et al. (2011) was used for improving the spatial resolution, as described in the next paragraph.

## 2.2 Emission estimation: satellite-only

The work of Pommier et al 2013, hereafter referred to as P13, served as starting point for our own analysis. A brief description of their method is given below. In P13 averages were made over respectively four and five years to analyse the concentrations change from period 1: 2000-2003 and period 2: 2004-2008 for eight large cities. In order to distinguish cities, besides the temporal averaging also spatial averaging was applied, using the spatial oversampling technique of Fioletov et al. (2011). For this satellite-only approach, a 200x200 km$^2$ area around the target city is mapped at 2x2 km$^2$ resolution, with each high-resolution grid cell representing the average value of all wind direction oriented satellite data having their footprint centre within 28 km distance of that cell. The pixels were rotated in the direction of the wind using the city centre as rotation point, to align the urban plumes in upwind-downwind direction. With this technique, the data were oversampled to prevent urban plumes of CO from being smoothed out during the spatiotemporal averaging, as described also in Streets et al. (2013). The difference between the average MOPITT retrieved upwind and downwind concentration was subsequently used as a proxy of emission strength. Further, the Relative Difference (RD) quantifies the relative change in the proxy of emission strength between the two time periods.

In our study, the same spatial averaging and wind rotation techniques were used. For the wind data, 3-hourly wind fields were used from the ERA Interim reanalysis project of the European Centre for Medium-Range Weather Forecasts (Berrisford et al., 2009). These fields were averaged at 1°x1° resolution and 60 hybrid sigma-pressure levels from the surface to the top of the atmosphere using the pre-processor that is used for generating wind fields for the global transport model TM5 (Krol et al., 2005). For each day, the wind direction was taken for the grid box in which the city centre of the respective city is located and the time step closest to the local overpass time of MOPITT. An average wind direction was constructed over the lowest 15 hybrid pressure layers of the TM5 model, roughly representing the average wind direction in the planetary boundary layer (PBL) up to about 750 hPa. For every MOPITT overpass, the associated modelled wind direction was recorded. This procedure is close but not identical to P13, who used 0.75°x0.75° data from ECMWF averaged from the surface to 700 hPa.

The urban concentration enhancement was finally estimated according to P13. First, for the total column CO, wind rotations and averages were made for the two periods. The emission proxy in molecules/cm$^2$ was then calculated as the difference between the average of the five maximum downwind total columns ($CO_{totdownwind}i$; molec/cm$^2$) minus the average of the





five minimum upwind CO total columns ($CO_{totupwind}i$; molec/cm$^2$) in a 20 km broad band from 100 km upwind to 100 km
downwind of the city in the respective period, $Vd - Vu$, according to Eq. 3 (from P13):

downwind-upwind difference= $V_d$ - V$_u$=

$$\frac{\sum_{i=1}^{5}\max(CO_{totdownwind}i)}{5} - \frac{\sum_{i=1}^{5}\min(CO_{totupwind}i)}{5} \quad (3)$$

The standard deviations of the 5 highest downwind and of the 5 lowest upwind concentrations were calculated. The sum
of these two standard deviations is used as the uncertainty in $Vd - Vu$. From $Vd - Vu$, the relative difference (RD) between
period 1 and period 2 was calculated to estimate the trend in the concentration enhancement. The RD is defined as the change
between the two periods with respect to period 1 and is expressed as a percentage.

### 2.3 Emission estimation: WRF optimization

To quantify emissions, additional information is required to determine the relation between emissions and concentrations,
involving transport. To take this into account, we combined the satellite data with model data from the Weather Research
and Forecast (WRF) model. We minimized the difference between the model and the satellite gridded data by changing the
emissions in WRF to find the most probable emissions. The method will be described in more detail in this section.

### 2.3.1 WRF model

Model simulations of CO over Madrid were performed using the WRF model (http://www.wrf-model.org/) version 3.2.1, with
the Advanced Research WRF core (ARW). WRF is a numerical non-hydrostatic model developed at the National Centers for
Environmental Prediction (NCEP). It has several choices of physical parameterizations, which allows application of the model
to a large range of spatial scales (Grell et al., 2005). For this study we used an updated version of the Yonsei University (YSU)
boundary layer scheme (Hu et al., 2013), the Unified Noah land surface model for surface physics (Ek et al., 2003; Tewari
et al., 2004), and the Dudhia scheme (Dudhia, 1989) and the Rapid Radiative Transfer Method (RRTM) for shortwave and
long wave radiation (Mlawer et al., 1997). Cloud physics are solved with the Grell-Freitas cumulus physics ensemble scheme
(Grell and Freitas, 2014). A built-in application of WRF-ARW is WRF-Chem (Grell et al., 2005), which deals with chemical
processes and tracer transport. WRF-Chem is an online model, which means that the tracer transport is consistent with all
conservative transport done by the meteorological model and that the chemistry can feedback on the dynamical computations.
In this research, only the model's tracer transport function was used, not the encoded chemistry of WRF, to speed up the
model. We considered this as a safe option, since the photochemical lifetime of CO is too long for its chemical degradation to
play a significant role during transport across the city domain. For our Madrid case study, we set the model's outer domain to
the Iberian Peninsula and part of the surrounding water bodies. This domain, modelled at a resolution of 30x30 km$^2$, defines
boundary conditions for a nested subdomain with a model resolution of 10x10 km$^2$ covering an area of 490x430 km$^2$ around
Madrid (Fig. 1). All the analyses in this paper were done for a sub region of 200x200 km$^2$ around Madrid within this second




domain. The time step used for calculations of dynamics and physics was 4 minutes in the outer domain and 80 seconds in the
inner domain. We used 30 dynamic vertical pressure levels between the surface and 50 hPa. The CO boundary conditions of the

outer domain were based on MOPITT climatological retrieved data. On each of the four lateral boundaries of the outer domain
of WRF, the 9 year (2000-2008) average MOPITT CO concentration per month is taken over a half-degree zone adjacent to
each boundary or the nearest land pixels of MOPITT. The data were interpolated to provide the vertical profile for all vertical
layers of WRF. These four, monthly varying, profiles have been implemented into WRF as lateral boundary conditions for
CO. This is considered sufficiently detailed, since the background concentrations will be scaled in our optimization technique

and no significant background pattern is expected to come with the data, which is also confirmed in section 3.5. The initial
concentrations of CO within the domains were set to zero and are expected to adapt quickly to the boundary conditions by
lateral transport. Initial and boundary conditions for meteorological parameters were based on data from the NCEP at a 1°x1°
spatial and 6-hourly temporal resolution.

### 2.3.2 Emission datasets

CO emissions to use as prior estimates were taken from different anthropogenic emission inventories that are available for
Madrid. For the years 2002 and 2006 we applied emissions from the EdgarV4.2 inventory (available at a resolution of 0.1°x0.1°)
for the corresponding years (Crippa et al., 2016). We also used emissions from the European TNO-MACC inventory (Kuenen
et al., 2014) with a spatial resolution of 0.125°x0.0625°, for the years 2006 (version III) and 2007 (version II). All the emissions
were re-gridded to the resolution of the WRF domains and account for monthly, weekly and hourly emission variations based

on temporal emissions factors reported by Gon van der et al. (2011). More information on the different sectors included in the
emission datasets can be found in Appendix A.

### 2.3.3 Comparing MOPITT and WRF

The information of the MOPITT retrievals is not equally distributed over the 10 vertical levels, as mentioned earlier. For a
fair comparison between satellite observations and model simulations, the **AK** matrix and a priori profile for each retrieval

has been applied to the corresponding model output, ensuring a consistent vertical weighting of the model compared with the
measurements. The MOPITT averaging kernel matrix was applied to the logarithm of model simulated CO concentrations
following Eq. 2, using the interpolated vertical model profile of CO from WRF as $x_{true}$, $x_{retr}$ forms then the WRF vertical
profile on MOPITT levels with the applied averaging kernel matrix that is used for comparison. In the comparison, average
mixing ratios over all vertical MOPITT layers are used. For this method we only used MOPITT V6 data.

### 2.3.4 Validation of WRF data

To verify the performance of the model, we compared the model simulated CO concentrations to available in-situ measure-
ments in Madrid (http://gestiona.madrid.org/azul_internet/html/web/InformAnalizadoresAccion.icm, accessed 19 December
2016). CO concentration data are available for 2006 from two locations within our WRF domain: Mostelos, a station in a park





in the south of Madrid and Villa del Prado, a background station in the Alberche basin. For both locations the concentrations and patterns in concentrations appear very similar between WRF and the observations, although WRF overestimates the concentrations at the Villa del Prado station (Fig. A1, upper panel). The variation over the months with higher concentrations in winter is well represented, most peaks seen in the observations are also found in the model and concentration differences between model and observation are generally within 0.1 mg/m$^3$. The overestimated CO concentration for the Villa del Prado station is considered reasonable, since with the resolution of 10x10 km$^2$ of WRF, the WRF pixel also includes two small towns in this area, while the station is measuring at a very remote location at the Villa del Prado station. On hourly time scale, WRF also follows the observations quite well (Fig. A2), stable low concentration patterns are also represented in the model as such and higher concentrations with morning and afternoon peaks are also represented, although WRF is not able to see all peaks and some peaks are under and overestimated (differences of up to 1 mg/m$^3$). Given the limited resolution used in WRF and the difficulty of representing measurement sites in an urban environment, we consider the performance of WRF adequate to make a reasonable comparison with the coarser resolution satellite data. For 2002, only data from the Mostelos station are available. In Fig.A1, lower panel, the comparison with these data is shown; the concentrations match also very reasonably for as well the peaks as the yearly patterns, the concentrations do most of the time overlap within 0.1 mg/m$^3$

### 2.3.5 Simulation period

To reduce the random noise and to increase the signal from relatively small sources, it is required to average MOPITT data over longer time periods as earlier studies already mentioned (e.g., Clerbaux et al. (2008); Girach and Nair (2014); Deeter et al. (2014)). Averaging times ranged in these studies from 1 month for the second study to 7 years for the first study; it should be noted, however, that these studies used coarser spatial resolutions. In our study we chose to average 1 year of data, which resulted in quite good comparison with WRF (R$^2$=0.75) and a clearly visible enhancement of CO mixing ratio over the city of Madrid. A description of the more detailed test we did that resulted in the use of a period of a year can be found in Appendix B.

### 2.3.6 From model mixing ratios to emissions

Several model simulations were done, i.e., with different emission datasets, for the years 2002 and 2006 for comparison with MOPITT. For each year also a background simulation without emissions was done, the boundary and initial conditions are kept the same as in the simulations with emission. The background simulations are done without any emissions: the CO in the data is only based on spreading from the boundaries, as well as with emissions outside of the 200x200 km$^2$ box around Madrid, but without emissions in the urban area where the optimization is performed. The difference between a simulation with and without urban emissions represented the contribution of the emissions of Madrid to the simulated CO concentrations. As is described in more detail in section 3.5, the emission optimizations gave comparable results for both backgrounds. Most of the results in this paper are therefore based on the simplest setup for the background simulation: the one without any emissions.

Since tracer transport in WRF is linear, the CO contribution from Madrid scales linearly with its emission. Because of this, the optimal, i.e., best fit, emission was linked to the inventory emission by a scaling factor ($f_{emis}$) of the simulated urban plume:





the difference between CO in the emission and background simulation. To make this method easily applicable to other regions and to limit the required WRF computation time, we implemented only direct anthropogenic CO emissions and assumed a

uniform distribution of other sources of CO (e.g., direct natural sources and indirect sources of CO such as the atmospheric oxidation of natural and anthropogenic volatile organic carbon compounds and methane from the city or the surrounding forests). To account for these missing sources in the domain, a background correction factor ($f_{back}$) was introduced that has no spatial pattern but is simply a multiplication factor of the concentrations in the background simulation.

After a WRF simulation, the WRF data were sampled according to the MOPITT retrievals, the **AK** matrix and MOPITT a

priori profile were applied, and the mixing ratios were gridded on a 2x2 km$^2$ grid and averaged over the entire column with the oversampling technique of Fioletov et al. (2011), as described in section 2.2 and used in P13. Taking the column value in molec/cm$^2$ as was done in P13 seemed to be less appropriate here, since the effect of orography would also be influencing the match between the model and satellite. Instead, the whole column average CO mixing ratio was taken to still maximize the available information.

To estimate CO emissions, we used a simple optimization scheme based on Brent's method (Brent, 1973; Press et al., 1992). We minimized the difference between MOPITT and WRF average column mixing ratios by varying $f_{backg}$ and $f_{emis}$ iteratively using Brent's method. Brent's method is a root finding algorithm, which we used to find the minimum of the quadratic cost function $J$ (ppb$^2$), defined in Eq. 4:

$$\boldsymbol{J} = \sum_{i=1}^{n} ((\boldsymbol{X}_{mod[i]}(f_{backg}, f_{emis}) - \boldsymbol{X}_{sat[i]})^2) \tag{4}$$

In this function, $n$ is the number of grid cells within the 200x200 km$^2$ optimization domain. $\boldsymbol{X}_{mod[i]}$ is the total column average mixing ratio (ppb) in the i$^{th}$ grid cell of the model and $\boldsymbol{X}_{sat[i]}$ the mixing ratio (ppb) in the corresponding MOPITT grid cell. To analyse the robustness of the method, we repeated the optimisation using different data filters to test the sensitivity to retrieval uncertainty, and investigated the effect of optimising the absolute difference instead of the quadratic difference in Eq. 4. Four different filtering criteria were used: 1) Filtering of MOPITT data that were more than three or 2) four standard

deviations from the yearly 200x200 km$^2$ mean MOPITT CO concentration, and filtering of data that were more than 3) three or 4) four standard deviations from the mean difference between WRF and MOPITT. The default procedure was to minimize quadratic differences and filter out differences of more than three times the standard deviation between WRF and MOPITT.

## 3 Results and discussion

### 3.1 Emission trend estimation and uncertainty based on satellite data only

The first method we used to estimate emission trends from large cities is the one applied before by P13. To estimate the uncertainty in these values, we used both version 5, as in P13, and version 6 of the MOPITT multispectral data in these calculations.





The typical downwind minus upwind MOPITT columns in our analysis - a proxy for the emission - range from $1 \times 10^{17}$ molecules/cm$^2$ (Madrid, Delhi, Paris) up to $7 \times 10^{17}$ molecules/cm$^2$ (Mexico City). When using MOPITT version 5 data (V5),

we found some significant differences between our study and P13 (total difference range: 0.006-1.8 $\times 10^{17}$ molec/cm$^2$), with an average discrepancy of $0.5 \times 10^{17}$ molecules/cm$^2$ (Table A2, Fig.A3).

The changes between the 2000-2003 and 2004-2008 periods, used to assess the trend in the emissions, are between $+0.2 \times 10^{17}$ and $-2.4 \times 10^{17}$ molecules/cm$^2$. This results in negative trends (RDs, see section 2.2) in the order of $-48\%$ to $-4\%$ for most cities (Fig. 2) and a positive RD of 15% for Delhi and +5% for Madrid. As we attempted to use exactly the same method

as P13, with only a slight difference in the use of wind data, our results suggest that the uncertainties of the emission proxies in P13 (0.01-0.1 $\times 10^{17}$ molecules/cm$^2$) were underestimated. A more realistic uncertainty for the emission proxy should rather be in the order of the mean discrepancy we found, i.e., $0.5 \times 10^{17}$ molecules/cm$^2$.

Comparisons of the MOPITT V6 data with P13, expected to give small differences due to the different retrieval algorithm of V6 compared to V5, also show rather large differences (Table A1), with an average discrepancy of $0.4 \times 10^{17}$ molecules/cm$^2$.

When the results of our approach are compared between using V5 and V6 of the data, we find discrepancies between $0.009 \times 10^{17}$ and $1.04 \times 10^{17}$ molecules/cm$^2$ with an average discrepancy of $0.3 \times 10^{17}$ molec/cm$^2$. The differences between V5 and V6 with our approach are thus smaller than the individual ones compared to P13, but still not negligible.

For Madrid, using V6, we find a negative trend of $-33\%$ (Table A1). The magnitudes of the RDs, see Fig. 2, found in our study are clearly different from those found in P13 and in the case of Sao Paulo the RD even shows an opposite sign (+40%

vs. $-27\%$ in P13). Using V6, only one of our RDs was within the error range of P13 given for the RD. For V5, only two of the RD estimations were inside the error range given in P13. The RD estimations, however, do agree with an absolute uncertainty of ~20% for most cities, so the method still has some value to make a rough estimation of trends in a very simple and fast way. An explanation for the large discrepancies in RDs, while the $V_d - V_u$ values are relatively close, is that the absolute changes between the two periods are close to our revised uncertainty estimate, and the RDs are thus almost in the uncertainty range of

the method.

Our results demonstrate that the method described in P13 gives a useful first guess of trends in emission, but also that the robustness of the method is only limited: the emission trends are small in comparison with the uncertainty in the upwind$-$downwind estimates and they are thus not well resolved by the method. V6 differs from V5 mainly by a correction for the geolocation bias, an updated a priori and different meteorological fields (Deeter, 2013a). In an attempt to better understand the factors limiting

the robustness of the approach, we identified a number of limitations inherent to the method, partly based on the differences between MOPITT V5 and V6, which will be discussed in the next section (3.2).

## 3.2 Limitations of the satellite-only approach

When using only satellite data to estimate emission trends, it is important to consider how satellite data are obtained: the maximum a posteriori retrieval is based on a set of measured radiances, a radiative transfer model, and a model-derived a priori profile. The averaging kernel represents the weighing of the measured signal and the a priori information in the retrieved CO profile (see section 2.1). In this section, we will analyse the possible influence of temporal variations in these terms on the





estimation of multi-year average emission trends from MOPITT retrievals, as well as the importance of the exact location of
the wind-turning centre. The effects of bias drift in the MOPITT retrievals, described in the validation papers (Deeter et al.,
2013, 2014), are not tested here. The influence is however, expected to be negligible, since the total column product is used
to estimate emission trends which has a drift of 0.001±0.003% per year for the V5 and 0.003 ±0.002% per year for the V6
multispectral product and the drift is existent in both the upwind and the downwind CO column.

### 3.2.1 Sampling differences and averaging period

The a priori information that is used in the MOPITT retrievals is the same each year, but accounts for seasonal variation. Close
to cities this seasonal variation reflects both the change in emissions over the year, with higher emissions in winter and low
emissions in summer and the seasonal cycle of the OH sink, which varies with season and peaks in summer (e.g. Girach and
Nair (2014); Lal et al. (2000); Novelli et al. (1998)), leading also to low CO mixing ratios in summer. Because of this, seasonal
variations in measurement coverage may bias annual averages. For example, a year with below average cloud cover during
summer - so less data filtered out - would lead to a lower annual average CO estimation compared to an average year, even if
the CO mixing ratios were exactly the same in those years. However, uneven sampling would not affect the RD calculation as
long as the background and the city signal are influenced equally. To investigate the sensitivity of the RD calculation to uneven
sampling, we analysed the a priori data for the years 2000-2008. The a priori is a good measure for this, since it is extracted
from the retrieval data and therefore sampled in the same way as the retrievals.

When we averaged a priori data, annual mean a priori CO varied by $1 \times 10^1 6$-$1 \times 10^1 7$ molec/cm$^2$ between years, which is of
the same order of magnitude as the long term trends in CO that are estimated with the satellite-only method (Fig. 3, left). The
effect can be seen very well in the years 2000 and 2001. In 2000 there are no satellite data for the months January and February,
biasing the average towards low summer columns. Oppositely, in 2001, June and July data are missing, which increases the
annual mean. In the right panel of Fig. 3, the downwind minus upwind concentration differences per year are calculated for the
a priori data for cities with enhanced CO mixing ratio over their centres in the a priori. For Baghdad, Moscow and Madrid, the
2000 $V_d - V_u$ is lower than that of 2001. New Delhi, with a different yearly CO pattern due to the monsoon, does not show
this difference. In this picture, however, also all the other years show varying emission proxys of similar quantity. This suggests
that the sampling problem has also a spatial dimension. The calculated RDs for the four cities based on a priori data are not zero
percent, as expected for annually repeating priors, but +11.8% for Madrid, $-13.3\%$ for Bagdad, 20.6% for Moscow and $-2\%$
for Delhi. These results indicate that temporal variations in sampling may significantly influence emissions trends obtained
using the satellite-only method.

Some recent studies on CO trends over larger regions overcame the uneven sampling problem by de-seasonalizing the data
before studying trends (Strode et al., 2016; Girach and Nair, 2014). In our method using the WRF model (see below), the
problem of uneven sampling is largely solved as we sample our model according to the availability of satellite data.



### 3.2.2 Role of the a priori

The a priori information of MOPITT version 6 is based on monthly climatologies, temporally and spatially interpolated to generate a priori values for a specific location and day (Deeter et al., 2014) on a 1°x1°(latitude x longitude) spatial resolution. This results in a priori fields which are already quite detailed: the a priori data of the eight cities of P13 and Madrid reveal

already the location of some of the large cities. The MOPITT V5 and V6 data make use of different a priori information. For all of the cities there are slightly different concentration patterns in the a priori products between these two versions. This raises the question to what extent the differences in emissions trends derived from the two MOPITT versions in Fig. 2 are explained by different a priori. To investigate this in more detail, we compared the emission estimation of the satellite-only approach for the standard and a uniform a priori over the whole domain. From this test, however, we could only find a minor contribution

of the a priori to the RD. For Madrid we find, for example, 2% change in RD estimation when a uniform a priori was used, for Baghdad we find a 3% change, for New Delhi a 6% change and for Moscow a 2% change. The differences are, however, somewhat larger, i.e. in the order of 5%, when we replace the version 6 a priori with the version 5 a priori data. This last step, however, required the use of the data that was available in both V6 and V5 of the data, leading to a decrease in the amount of data where the estimations were based on. To be sure to look at the effect of the a priori only, we used the WRF model data for

the years 2002 and 2006 to calculate the RD with a uniform a priori (the average MOPITT a priori) and the standard MOPITT a priori. From this test, we found a decrease in the RD of only 1.2% when the uniform a priori was used. The change in a priori thus causes around 5% change in RD estimation between version 5 and 6.

### 3.2.3 Averaging kernel stability

Since the city CO emissions take place in the lowest layers of the atmosphere, the amplitude of the retrieved city signal depends

strongly on the sensitivity of the MOPITT retrieval to these altitudes; any temporal change in this sensitivity will influence the emission trend estimation. Yoon et al. (2013) already concluded that a temporal change in the **AK** can lead to a significant error in the trend estimation of retrieved CO. Our analysis shows that there is a change in the average multispectral **AK** shape over the years 2000 to 2004 over Madrid (Fig. 4). The slight shift in **AK** sensitivity reduces the sensitivity to the lowest layers (surface to 800 hPa) and increases the sensitivity to the mid-troposphere (300-500 hPa). After 2004, these sensitivities stabilize,

except for some year-to-year variation. To show this, we used the **AK** area (Rodgers, 2000): for each vertical layer the sum of all values of the corresponding row in the **AK** matrix, averaged over the years for all months with data in all years of our sample period (i.e. March-December, except June, July); note that the figures are very similar to the figures where all available months are taken for each year (not shown). We found downward trends near the surface of $-16 \pm 6$ %, and upward trends at 400 hPa of $+8 \pm 3$ % over the years 2000-2004 (Fig.4, right panel). In Fig.4 we show this effect for Madrid, but it is visible for all

cities analysed in P13. This sensitivity change might have been caused by instrument degradation, variability in meteorological conditions and/or changes in the CO abundance over the years (Strode et al., 2016). The NIR data show a decreasing sensitivity over all layers in time. The increasing sensitivity to the layers higher up comes from the TIR data (Fig.A4).





The **AK** trends may not be large but the city CO signal compared to background is not large either. As the CO concentration gradient around sources is largest in the layers near the surface, and lower higher up, the trend in the **AK** causes an artificial negative trend in the concentration enhancement over cities, biasing the emission trends derived from the satellite-only method. For Madrid, we tested this by constructing a synthetic dataset of MOPITT retrievals for the years 2000 to 2008, all based on WRF-Chem simulated CO vertical profiles over Madrid for 2002. For each year, every **AK** is scaled such that the annual mean

sensitivity remains at the level of 2002 for each **AK** layer. This led to a negative RD of −5%. From this result, we conclude that the stability of the **AK** is influencing the emission trend estimation using the satellite-only method, which introduces an uncertainty when using satellite data from MOPITT and potentially also other instruments. It should be noted, however, that the averaging kernel is quite specific for each retrieval and replacing it by a corrected **AK**, as done here, is justified as a sensitivity test but is not considered a solution to the problem, as indicated by the data description paper published in Deeter (2002).

### 3.2.4 the rotation point selection

In the satellite-only approach, a wind rotation technique is applied to calculate upwind − downwind differences. This technique selects a single point in the centre of the city as rotation point. However, we found that the estimated upwind − downwind differences are sensitive to the location of this rotation point, which is problematic since it is hard to tell what the exact centre of a city is. Moving this rotation point for example from the centre defined by Wikipedia to the centre point defined by Google

Maps (GM), which differs 0.7-3.9 km for our selected cities - both locations could be equally well defined as centre - gives downwind−upwind differences varying by $0.03 \times 10^{17}$-$0.3 \times 10^{17}$ molec/cm$^2$, corresponding to RDs varying by 8%-25% (Fig.5). As a solution for this problem, we using the weighted emission centre of the city instead of the general centre would be a fairer way to use this method. We tested this for the city of Madrid for the weighted centre point in the TNO-MACC emission inventory and weighted centre point of the EdgarV4.2 emission inventory. We found a positive RD of +3% for the Edgar

centre and a negative RD of −4% for the MACC centre, which was located 8 km more southwards. These last estimations are probably better estimations of the real trend, since it uses the centre of the emissions instead of the centre of the buildings, but it also shows that this problem is difficult to solve, since the exact centre of emissions is also not known.

The satellite-only method is thus highly sensitive to the selected location of the rotation point, which introduces a large uncertainty in the estimated emission trends. This outcome is particularly relevant for the use of MOPITT data, because of a

location bias in MOPITT version 5, which has been corrected in version 6. This can be an important reason for the differences in emission trends found between V5 and V6. The geolocation bias correction that was used in P13 and our study was slightly different from the correction done for V6 of the data by the MOPITT team (Deeter, 2012). As we saw in this paragraph only a small shift in the location already can change the RD estimation substantially.

### 3.3 Emission estimation based on WRF optimization method

To overcome the limitations of the satellite-only approach and to be able to quantify emissions, we developed a different method using the WRF model in addition to the satellite data. For this method, the model is sampled at the location and time of each individual satellite measurement. Since the model accounts for the seasonality in CO, the model and satellite data are influenced





in the same way by uneven seasonal sampling. Therefore, its influence on the derived trend is expected to cancel out. The model optimization approach does not need wind rotation, avoiding the uncertainties introduced by this procedure. Likewise, any variation or trend in the **AK** influences the model in the same way as it does with the measurements. In addition, the model accounts for influences of varying meteorological conditions on the dispersion of the city plume. Besides these advantages of using WRF, there is one notable drawback, which is the computational cost of a simulation covering several years. As explained in the methods section, we do simulations of 1 year; the accompanying $R^2$ between the gridded oversampled WRF

and MOPITT is then 0.75.

Emissions were estimated by minimizing the cost function as described in the methods section (see paragraph 2.3.6). In all simulations, the modelled CO columns were smaller over the whole domain compared to the satellite, probably due to the omission of secondary and natural CO sources (e.g. from oxidation of naturally emitted hydrocarbons) in the model. Over larger geographical regions, biogenic sources can contribute to 40%-80% of the CO column (Choi et al., 2010; Hudman et al.,

2008). As explained, we therefore optimize both the background and the anthropogenic emissions by two scaling factors, taking into account the **AK** in the comparison between MOPITT and the WRF data. We performed emission optimizations for the years 2002 and 2006. Starting with the initial emissions for each associated year from EdgarV4.2, we find optimum of 52% of the EdgarV4.2 emissions in 2002 and 83% of the estimated EdgarV4.2 emissions in 2006. This allows us to directly estimate emissions for Madrid for these years: averaged over the 200x200 km$^2$ domain the corresponding emission is 0.22 Tg of CO

for 2002 and 0.20 Tg of CO for 2006. Fig. 6 and 7 show the column averaged mixing ratio patterns before and after optimizing the emission, in comparison with the MOPITT signal and the remaining difference between WRF and MOPITT.

Fig. A5 shows for the years 2002 and 2006 the offset between the model and the satellite data before and after applying the background and emission optimisation. The initial misfits are in the range of 0 to −8 ppb (around 4% relative to the mean CO column mixing ratio around Madrid of ~90 ppb). The model gives initially lower concentrations than the satellite, which is

accounted for in the optimization of the background.

It must be noted, however, that our method is quite sensitive to specific settings used in the inversion. To further investigate the robustness of the WRF optimization method a series of sensitivity experiments have been performed, varying the data filtering method (section 2.3.6) and the a priori emissions (using EdgarV4.2, TNO-MACC-II and TNO-MACC-III). The results of these tests are summarized in Fig. 8 and Table 1. When we average the results of all tests, the average optimum is 45% of the

original emission for 2002 and 87% of the original emission for 2006 (Fig. 9, upper panel). This is quite close to the estimates from the standard method, although the range of possible emissions indicates a sizeable uncertainty: for 2002, the emissions range between 0.15 and 0.24 Tg of CO over the 200x200 km$^2$ area around the city centre of Madrid, for 2006 this range is between 0.19 and 0.26 (with one outlier of 0.32) Tg CO, an uncertainty of 23% on the average value. Including the TNO-MACC (versions 2 and 3, for the year 2006) inventories as alternative emission patterns, upper part of the range increased to

0.44 Tg for 2006 (Fig. 9, lower panel), based on the new average value, this is an uncertainty of 56%. The large sensitivity to the a priori emission pattern can be explained by the use of a single scaling factor to optimize the city emissions. Therefore, uncertainties in the emission inventory pattern, for example due to missing sources, are difficult to correct for, using our current



inverse modelling setup. This was found to be a more general problem in inversion studies (see the recent publication by Jacob
et al. (2016)). In section 3.5, we describe some steps that we took to solve this problem in more detail.



**Table 1.** Optimization-derived CO emissions comparing different approaches

| Emission inventory | Background run | Optimization method | | 2002 emission [kg/yr] | 2006 emission [kg/yr] |
|---|---|---|---|---|---|
| Edgar emissions | No anthropogenic emissions outside 200x200km$^2$ | abs(y1−y2) | No filter | 2.31E+08 | 1.97E+08 |
| | | | Filter >3 stdev difference WRF−MOPITT | 2.31E+08 | 1.98E+08 |
| | | (y1−y2)$^2$ | No filter | 2.00E+08 | 1.97E+08 |
| | | | Filter >3 stdev difference WRF−MOPITT | 2.02E+08 | 1.99E+08 |
| | | | Filter >3 stdev difference squared | 2.15E+08 | 1.97E+08 |
| | | | Filter MOITT > 4x stdev outliers | 2.00E+08 | 2.57E+08 |
| | | | Filter MOPITT >3x stdev outliers | 2.40E+08 | 2.34E+08 |
| | | 20x20 | No filter | 1.95E+08 | 2.02E+08 |
| | | | Filter >3 stdev difference WRF−MOPITT | 1.95E+08 | 2.03E+08 |
| Edgar emissions | Anthropogenic emissions outside 200x200 km$^2$ | abs(y1−y2) | No filter | 2.17E+08 | 1.92E+08 |
| | | | Filter >3 stdev difference WRF − MOPITT | 2.16E+08 | 1.93E+08 |
| | | (y1−y2)$^2$ | No filter | 1.58E+08 | 1.90E+08 |
| | | | Filter >3 stdev difference WRF−MOPITT | 1.62E+08 | 1.91E+08 |
| | | | Filter >3 stdev difference squared | 1.84E+08 | 1.90E+08 |
| | | | Filter MOITT > 4x stdev outliers | 1.58E+08 | 2.46E+08 |
| | | | Filter MOPITT >3x stdev outliers | 1.93E+08 | 3.19E+08 |
| | | 20x20 | No filter | 1.55E+08 | 1.91E+08 |
| | | | Filter >3 stdev difference WRF−MOPITT | 1.56E+08 | 1.92E+08 |
| MACCv3 emissions | No anthropogenic emissions outside 200x200km$^2$ | abs(y1−y2) | No filter | | 3.59E+08 |
| | | | Filter >3 stdev difference WRF−MOPITT | | 3.58E+08 |
| | | (y1−y2)$^2$ | No filter | | 3.75E+08 |
| | | | Filter >3 stdev difference WRF−MOPITT | | 3.74E+08 |



### 3.4 Trend estimation with the WRF optimization method

To infer the trend in CO emissions from Madrid using the WRF optimization method, emissions were optimized for two different years: 2002 and 2006. Because of the three years in between and the limited inter-annual variability, it is possible to estimate the trend in emissions over Madrid in this period. Both the EdgarV4.2 and the TNO-MACC-III emission inventories report downward trends in the emissions over Madrid, with EdgarV4.2 showing the largest decrease ($-46\%$ and $-25\%$ for respectively EdgarV4.2 and TNO-MACC-III between 2002 and 2006 over Madrid). With our emission optimization approach, however, we found a trend of only $-8\%$. Averaged over all sensitivity tests, we even found an upward trend of about 8% (Fig. 9, upper panel). When the TNO-MACC II or III emissions were used to simulate the city plume we find a 35% increase in emission between 2002 and 2006 (Fig. 9, lower panel).

In the satellite-only approach, as mentioned earlier, we find for V6 a decrease of 32% between the 2000-2003 and the 2004-2008 period over Madrid. However, when we limit this satellite-only analysis to the years 2002 and 2006, a 5% emission increase is found ($V_d - V_u = 1.01 \times 10^{17}$ in 2002 and $1.07 \times 10^{17}$ in 2006), which is in better agreement with the increase estimated using the WRF optimization method.

In all cases, the emission estimation and trend seem to be lower and less negative than emission and trend reported by EdgarV4.2 over Madrid and more similar to the TNO-MACC-III inventory.

### 3.5 Limitations of the WRF optimization method

As described in the previous paragraphs, the optimization method combining MOPITT retrievals and WRF model output has advantages over the satellite-only approach, but comes with its own limitations and uncertainties.

An important source of uncertainty is the background optimization. As can be seen in the images in the right most columns of Fig. 6 and 7, considerable differences between MOPITT and WRF remain in the background column mean mixing ratios after optimization. Optimizing the background with a single scaling factor for the whole domain is clearly insufficient to account for the complex pattern of differences between the model and the satellite.

Part of the pattern is probably still related to noise in the MOPITT data, since we did not filter for very low or high values in MOPITT, although they can have an important effect on several cells with the oversampling technique. We performed an additional optimization in which we reduced the spatial resolution by averaging the retrievals and model data to a 20x20 km$^2$ grid (instead of 2x2 km$^2$) in the domain around Madrid. Using this approach, we find reduced optimal emissions, with differences up to 20% (Table 1, optimization method: 20x20).

Another possible explanation for the remaining differences between the modelled and observed patterns might be other sources of CO, which are not (yet) included in the WRF model, such as the atmospheric oxidation of volatile organic carbon compounds from the city or the surrounding forests. Some forested areas in the north of Madrid indeed appear to be blue on the difference maps of both 2002 and 2006, pointing to underestimated concentrations in the model compared to MOPITT, suggesting that emissions of short-lived biogenic volatile organic carbon (VOC, quickly converted to CO) emitted from forests might play a role.



Finally, an important factor limiting the robustness of the WRF optimization method is the prior emission pattern used in WRF for Madrid. This factor has been investigated in further detail by (1) changing WRF's background emissions, (2) inspecting the differences when using a different emission pattern by using both TNO-MACC and EdgarV4.2 emissions as priori in the model for 2006, (3) using the EdgarV4.2 2006 emissions as prior in the model for 2002 and (4) using TIR instead of the multispectral MOPITT data to do the optimization. The results have been analysed by examining the impact on the shape

of the cost function (Fig. 8). While the value of the cost function at the minimum quantifies how well the data are fitted, the second derivative of the cost function quantifies the robustness of the emission estimate. For all the 2006 optimizations the second derivative of the cost function is lower, i.e., is less steep than for the standard optimization for 2002, indicating that the uncertainty of the estimated emissions is smaller for 2002 than for 2006. The effect of the different sensitivity tests on the cost function is described below.

To investigate the contribution of emissions outside the optimisation area on the pattern in CO in the optimisation area, we performed a sensitivity test (sensitivity 1) by replacing the normal background simulation, without emissions, with a background simulation that has emissions in the area outside the optimisation area (see section 2.3.6). In the ideal case the background emissions only contribute to the background of the 200x200 $km^2$ area around Madrid without a pattern, so the method we used now to optimize the background with only one factor is able to account for this. If the emissions do contribute to the pattern, we expect the results to have lower cost function values in the optimum. The impact on the optimized emission

of Madrid was, however, well within the estimation uncertainty, as can be seen in Fig. 8 from the difference between solid and dotted lines. These show that the differences between the cost function values with and without accounting for these emissions are negligible. The emission estimates, however, with this replaced background, are, especially for 2002 consistently lower than with the standard background, on average 16% for 2002 and 1% for 2006.

Emission patterns differ between the TNO-MACC and the Edgar inventories (sensitivity 2). The cost function minimum was slightly lower for the simulation with the TNO-MACC-III inventory compared to the simulation that uses Edgar emissions. The TNO-MACC-III simulation, however, also produces a minimum that is clearly less confined and therefore less robust.

For 2002, implementing 2002 emissions clearly gave better results than implementing 2006 emissions (sensitivity 3, not shown). In the end, the most reliable results for 2002 and 2006 were obtained using EdgarV4.2 emissions in combination with

multispectral data.

The cost function of the TIR optimization (sensitivity 4) is as steep as that of the standard multispectral optimization, but the cost function values are much higher in the minimum, indicating that the TIR data are more difficult to fit by scaling the emissions in WRF. This can be explained by emissions outside the 200x200$km^2$ region having a relatively strong influence on the CO mixing ratios at altitudes where the TIR retrievals are most sensitive.

Despite the various influences on the accuracy of the WRF optimization discussed in this section, the uncertainties in the estimates, 23% for 2002 and up to 56% for 2006 are still smaller than the reported uncertainties in the emission inventories of 50%-200%. This confirms that estimating city CO emissions using MOPITT and WRF seems feasible. However, the current noise in MOPITT data requires averaging over at least yearly time periods before there was a clearly distinguishable signal of Madrid. Next to this, further improvements in the methodology are needed to decrease the uncertainty, such as the improved





treatment of the background concentration. It should also be noted that we did not test for errors in WRF in the representation of the dilution and advection apart from the comparison we made with local ground measurements (section 2.3.4).

## 4 Summary and conclusions

We have developed a new method to quantify CO emissions of cities based on a combination of satellite data and model simulations. This method is an extension of the method developed by Pommier et al. (2013), based on the pixel averaging

technique of Fioletov et al. 2011 to oversample satellite data, enabling the city signals to be distinguished within a reasonable time frame. We extended the urban-scale emission trend estimation techniques by adding CO mole fractions modelled with the WRF model. The comparison of model and satellite data enabled us to quantify the CO emissions over Madrid, whereas the satellite-only method was only able to determine a trend in the emissions. We identified and discussed limitations of the satellite-only technique: it is influenced by sampling differences between years, it is slightly dependent on the a priori

information used in the MOPITT retrievals (RD changes ~3%-5%), it is influenced by a trend in the averaging kernel (RD changes 5%) and it is strongly dependent on the exact location of the wind-rotation (RD changes up to 25% for locations up to 5 kilometres apart). Our results suggest that the uncertainties of the emission proxies in P13 ($0.01$-$0.1 \times 10^{17}$ molecules/cm$^2$) are too optimistic. A more realistic uncertainty for the emission proxy should rather be in the order of the mean discrepancy that we found between our results for V5 of the MOPITT data and P13, i.e., $0.5 \times 10^{17}$ molecules/cm$^2$. The absolute changes

between the two periods in emission proxy are close to our revised uncertainty estimate. This leads to RDs that are very often in the uncertainty range of the method.

Some effort can be made to overcome the largest part of these problems, by e.g., deseasonalizing the data, accounting for the change in **AK** and using the emission inventory centre for wind rotation of the data. This will probably increase the reliability and robustness of the satellite-only trend estimation. We chose, however, to investigate another method, which also enabled us to quantify the emissions. With this method, we do not suffer from the limitations of the satellite-only approach, as in our approach the model data is sampled according to the satellite data and no wind rotation is required because the model accounts for influences of varying meteorological conditions on the dispersion of the city plume. For the WRF-optimization method, it

5 is needed to average one year of data to sufficiently reduce the noise in the MOPITT retrievals to observe a clear signal from the city of Madrid. Averaging over a year will also smooth both the MOPITT and WRF data and reduce the effect of random model errors, while still providing a shorter period compared to the four and five year periods used in P13. To estimate the emissions, a quadratic cost function of the difference between the satellite and model data was minimized by adapting the emissions in the model. The optimum was found using Brent's method scaling two factors. To account for missing sources, we

10 optimized the background concentrations with a single scaling factor over the whole area. The emission estimation is based on the change in emission factor.

For 2002 we found that at the optimum the emissions were 0.52 times the original emissions in Edgar. For 2006 we estimated the emissions to be 0.83 times the reported emissions in Edgar. These values are more in agreement with the TNO-MACC-III inventory values for emissions around Madrid. After optimization, however, the remaining differences between WRF and





MOPITT are still large. This is probably caused by differences in the CO patterns between MOPITT and WRF, especially

for 2006. Additional data filtering to reduce this error or the use of other a priori emission patterns influences the optimized emissions significantly. For 2002 we found a possible range of emissions between 0.15 and 0.24 Tg of CO over the 200x200 km2 area around the city centre of Madrid, for 2006 the estimations range between 0.19 and 0.26 (with one outlier of 0.32) Tg CO. Or, expressed as a percentage, an uncertainty of 23% in the 2002 emission and up to 56% for the 2006 emission. These values are still smaller than the reported uncertainties in the used emission inventories of 50%-200% (Kuenen et al., 2014).

These uncertainties are comparable to our estimated uncertainty in the satellite-only method, but we also note that this new method is able to quantify emissions and that the uncertainties are based on one-year average MOPITT and model data, instead of the 4 and 5 year averages which were used in the satellite-only method. Our relatively simple method can thus be used to make an (approximate) estimation of city emissions. Our study confirms that estimating city CO emissions using MOPITT and WRF is feasible, however, further development of the method is needed to improve precision and robustness.



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



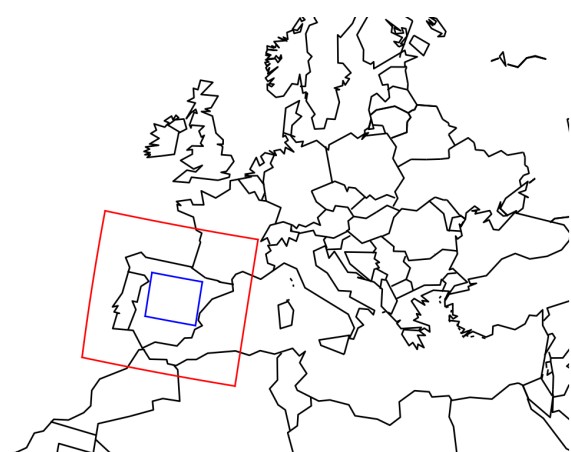

**Figure 1.** WRF domains d01 (red, 1500km x 1440km, resolution: 30x30 km$^2$) and d02 (blue, 490km x 430 km, resolution: 10x10 km$^2$).

Tewari, M., Chen, F., Wang, W., Dudhia, J., LeMone, M. A., Mitchell, K., Ek, M., Gayno, G., Wegiel, J., and Cuenca, R. H.: Implementation and verification of the unified Noah land surface model in the WRF model. , in: th Conference on Weather Analysis and Forecasting 11-15 January 2004, pp. 1–6, Seattle, 2004.

WHO, W. H. O.: Environmental health criteria 213: Carbon monoxide, Tech. rep., 2004.

Worden, H. M., Deeter, M. N., Edwards, D. P., Gille, J. C., Drummond, J. R., and Nédélec, P.: Observations of near-surface carbon monoxide from space using MOPITT multispectral retrievals, Journal of Geophysical Research, 115, D18 314, 2010.

Worden, H. M., Cheng, Y., and Pfister, G.: Satellite-based estimates of reduced CO and CO2 emissions due to traffic restrictions during the
5     2008 Beijing Olympics, Geophys. Res. Lett., 2012.

Yin, Y., Chevallier, F., Ciais, P., Broquet, G., Fortems-Cheiney, A., Pison, I., and Saunois, M.: Decadal trends in global CO emissions as seen by MOPITT, Atmospheric Chemistry and Physics, 15, 13 433–13 451, 2015.

Yoon, J., Pozzer, A., Hoor, P., Chang, D. Y., Beirle, S., Wagner, T., Schloegl, S., Lelieveld, J., and Worden, H. M.: Technical Note: Temporal change in averaging kernels as a source of uncertainty in trend estimates of carbon monoxide retrieved from MOPITT, Atmospheric
10     Chemistry and Physics, 13, 11 307–11 316, 2013.




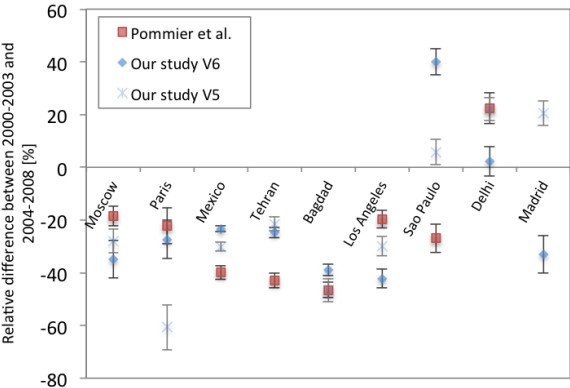

**Figure 2.** Calculated Relative Differences, comparing results of the satellite-only approach from this study (diamonds for MOPITT version 6, stars for MOPITT version 5) and the study of Pommier et al. (2013; squares). The error bars represent trend uncertainties, following the calculation method that was used in P13





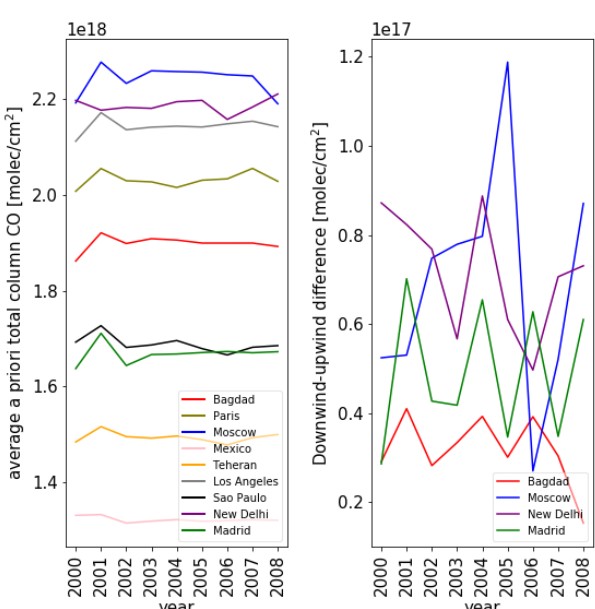

**Figure 3.** Left: variations in annual mean a priori total column CO over the years due to uneven sampling. Averages were made over the 200x200 km$^2$ domain around each city. Right: variations in annual mean downwind−upwind differences in total column a priori CO over the years, only cities with a distinct city-like pattern in the a priori are shown.





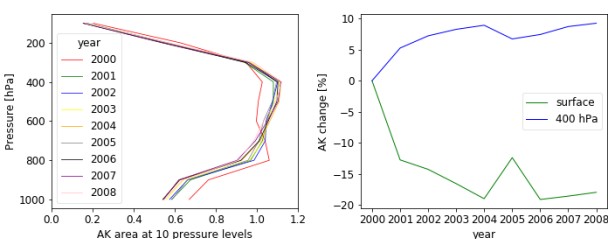

**Figure 4.** Yearly averaged **AK** area (Rodgers, 2000) values over the 400km$^2$ area around Madrid for the years 2000 to 2008, March - December (except June, July to minimize biases from uneven sampling), for the V6 NIRTIR product. Left: vertical profiles from the surface to the top level for corresponding main diagonal value of the **AK**. Right: change in average **AK** compared to the year 2000 for the surface level (blue) and 400hPa level (green).



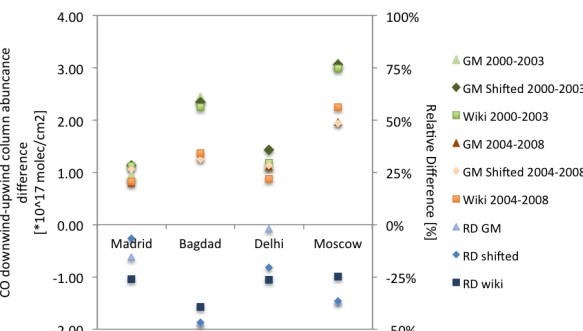

**Figure 5.** Upwind − Downwind difference (left axis, orange, green) and Relative Difference calculation (right axis, blue points) for Madrid, Bagdad, Delhi and Moscow using different rotation points within the city centre. GM: GoogleMaps location of the centre, GM shifted: 5 km shift of this point to another center location, Wiki: Wikipedia location of the centre. Wikipedia centre points are off by 3.9, 3.1, 2.1 and 0.7 km from the GM centre points for Madrid, Bagdad, Delhi and Moscow respectively.





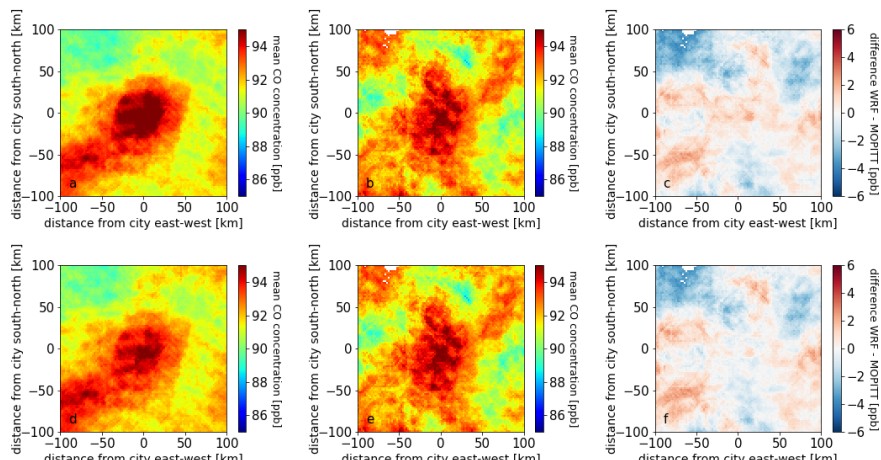

**Figure 6.** Column average mixing ratios of CO for 2002 before and after emission optimization in WRF: a) only background optimization.

b) MOPITT V6 signal. c) Difference WRF−MOPITT after background optimization. d) WRF after background and emission optimization.

e) As b. f) As c but now after background and emission optimization. The optimal emission is found to be 0.52 times the original emission.



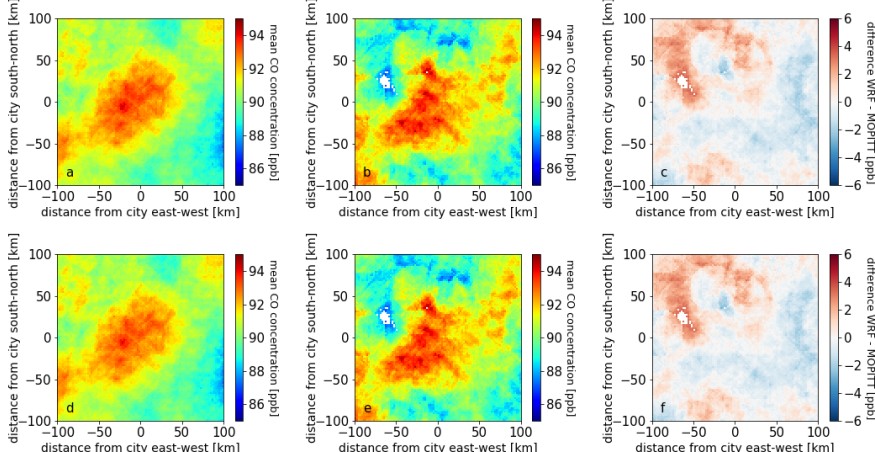

**Figure 7.** As Fig. 6 for 2006: (a) only background optimization. (b) MOPITT V6 signal. (c) Difference WRF−MOPITT after background optimization. (d) WRF after background and emission optimization. (e) As b. (f) As c but now after background and emission optimization. The optimum emission is found to be 0.83 times the original emission

Differences with the emission inventories of this magnitude are very well possible: the EMEP/EEA air pollution guide, also referenced in the articles describing the TNO-MACC emission dataset, reports uncertainties for CO emissions in the range of 50 and 200% for the sources that are most important in cities, such as (road) transport and commercial, institutional and residential combustion (European Environment Agency, 2013).





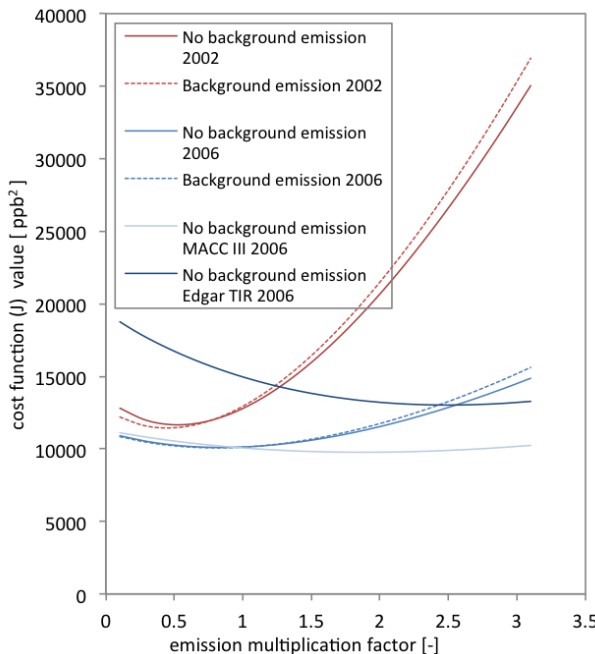

**Figure 8.** Comparison of the cost functions of WRF inversions using Edgar for the year 2002 (red), 2006 (blue), MACC III for 2006 (light blue). Dark blue: Inversion using Edgar and MOPITT V6 TIR data instead of NIRTIR for 2006. Dotted lines: Emissions outside the $200x200km^2$ area are accounted for in the background run. Solid lines: No emissions outside the $200x200km^2$ area in the background run. Note that for the MACC run the initial emission is lower than for the Edgar run, so the multiplication factor does not give an indication of the quantitative difference in optimal emission.



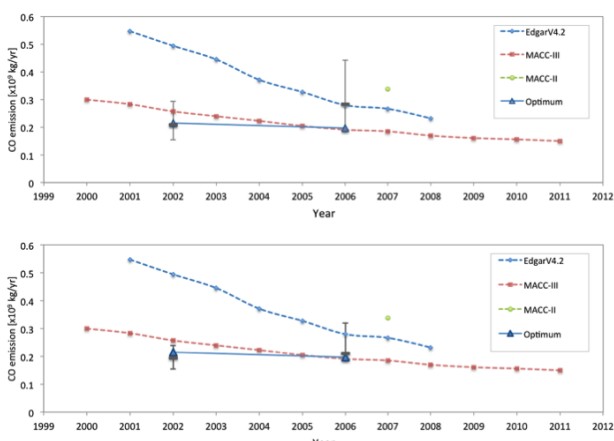

**Figure 9.** CO emissions in totals per year for the 200x200 km² area around Madrid, comparing inversion and inventory estimates. Blue triangles, solid line: inversion results for the year 2002 and 2006; blue dotted: EdgarV4.2; Green: TNO-MACC-II; Red dotted: TNO-MACC-III. The grey error bars and thick grey bar represent the range and the mean of the solutions obtained in various sensitivity tests (see text). The uncertainty of the Edgar and MACC emission inventory estimates are estimated at 50%-200% (Kuenen et al., 2014).



## Appendix A:  Appendix A: Emission datasets

Sectors in Edgar: Agricultural waste burning, residential, road transportation, non-road transportation, fossil fuel fires, large scale biomass burning (Emissions from savannah burning (4E) and land use change and forestry (5) are not gridded), combustion in manufacturing industry, metal processes, energy industry and waste incinerator, non-metallic paper chemical industry; transformation, oil production and refinering.

Sectors in MACC: Combustion in energy and transformation industries, non-industrial combustion plants, combustion in manufacturing industry, production processes, extraction and distribution of fossil fuels and geothermal energy, solvents and other product use, road transport, other mobile sources and machinery, waste treatment and disposal, agriculture.

## Appendix B:  Appendix B: Simulation periods

For the quantification of CO emissions from Madrid, we tested four different simulation periods in WRF. In this test, we optimized the trade-off between minimizing model calculation time and maximizing retrieval information content. The following averaging periods were selected: 10 days (from 1-10 July 2006), a full month (July 2006), a four months summer season (June-September 2006, JJAS) and a full year (2006). The shorter periods are all chosen in summer, as most data are available in this season. WRF was sampled for each individual MOPITT retrieval applying the AK, as described earlier, and a spatial comparison was made between the WRF and MOPITT-derived images of $200 \times 200 \text{km}^2$ over Madrid. For each period the oversampling method was applied to grid the data on this $2 \times 2 \text{km}^2$ grid. The scatterplots of these gridded data are shown in Fig. A6. Each subplot consists of the 10,000 points of this grid (note that for the shorter periods, there are overlapping points, originating from neighbouring grid cells that rely on the same data). Generally, the spatial variation in the WRF column averaged CO mixing ratios is much smaller compared to the MOPITT data, because of the limited precision of the individual data and the smaller variability in the CO signal in WRF. After averaging 10 days and 1 month of data the variability in MOPITT is still much higher than the variability in WRF, R2 values are respectively 0.43 and 0.33. This is probably partly due to the high measurement noise in MOPITT and partly caused by the stability of the model. Using four summer months (JJAS) or one year leads to better results, with R2 values of 0.55 and 0.75 respectively. The period of a year gave clearly the best, and useful, results and was therefore selected for emission estimation. A CO mixing ratio enhancement over the city was also best visible for the yearly period (not shown). Earlier studies already mentioned the need of averaging MOPITT data over longer periods to reduce the random noise and to increase the signal from sources (e.g., Clerbaux et al. (2008); Girach and Nair (2014); Deeter et al. (2014)). Averaging times ranged from 1 month for the second study to 7 years for the first study. It should be noted that these studies used coarser spatial resolutions.





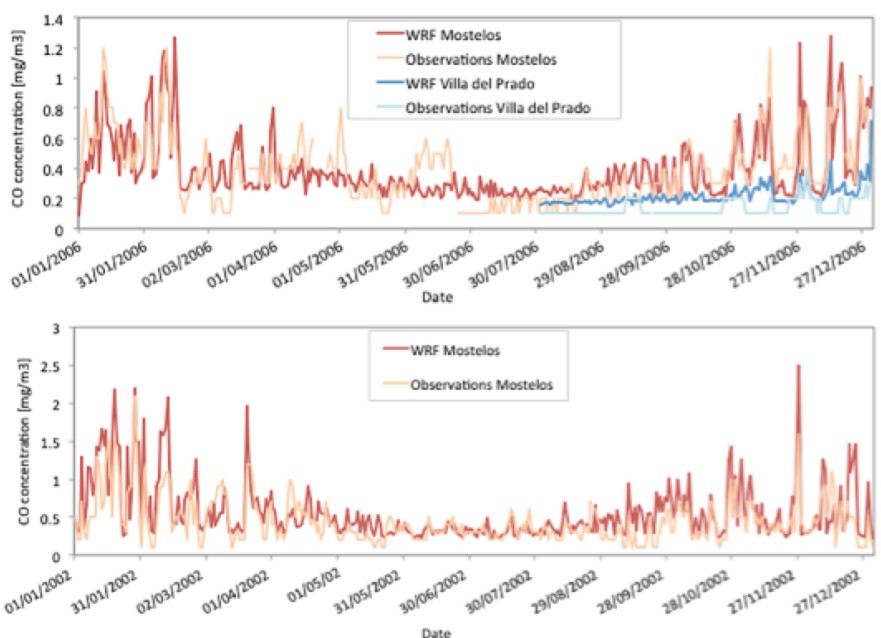

**Figure A1.** For 2006, above and 2002, below: daily averaged WRF surface concentrations (solid lines) compared to observations (dotted lines) at two locations near Madrid, the background and emission correction factors for each location found by our emission optimization method are applied.





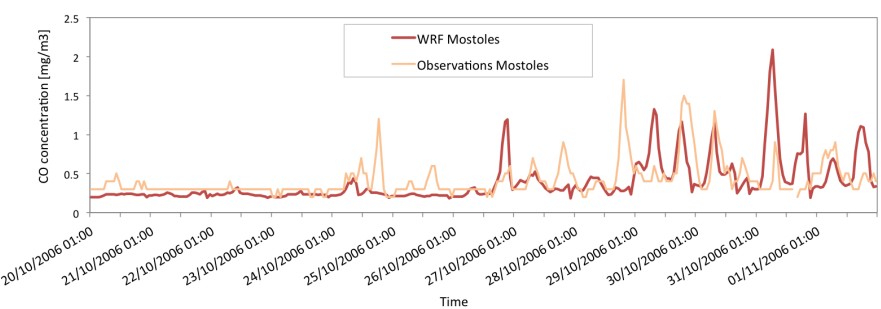

**Figure A2.** Hourly WRF surface concentrations (solid lines) compared to observations (dotted lines) at two locations near Madrid for 10 days in October. The background and emission correction factors for each location found by our emission optimization method are applied.





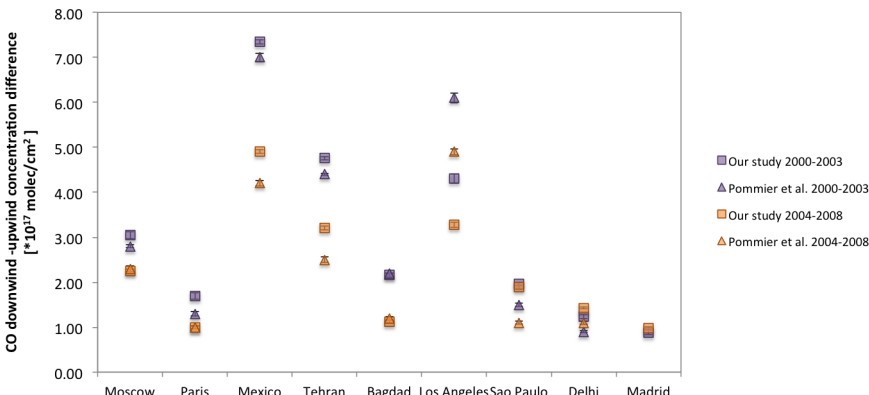

**Figure A3.** Total column CO concentration downwind minus upwind of selected cities (see methods-section), comparing our study using MOPITT version 5 (squares) and the study of Pommier et al. (2013, triangles). Error bars represent uncertainties calculated according to P13.





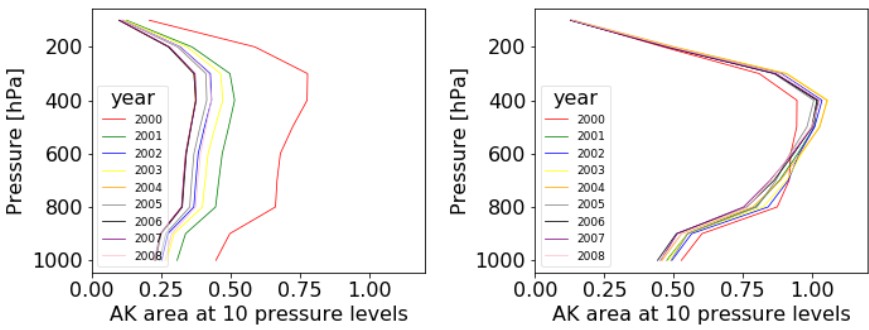

**Figure A4.** Yearly averaged **AK** area (Rodgers, 2000) values for the 200x200km$^2$ domain around Madrid from the surface (values plotted at 1000hPa, note that the average surface pressure around Madrid is actually closer to 900hPa) to the 50 hPa level for the years 2000-2008, March to December (except June, July to minimize biases from uneven sampling, for NIR (left) and TIR (right).





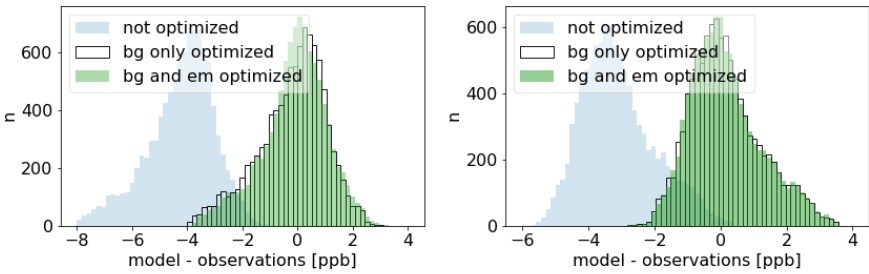

**Figure A5.** Comparison of prior and posterior misfits of the WRF model to the MOPITT retrievals. Left: year 2002, right: year 2006. Blue bars depicture the difference between the model and satellite data before optimization, the white bars the difference after background optimization and the green bars the difference after background and emission optimization.





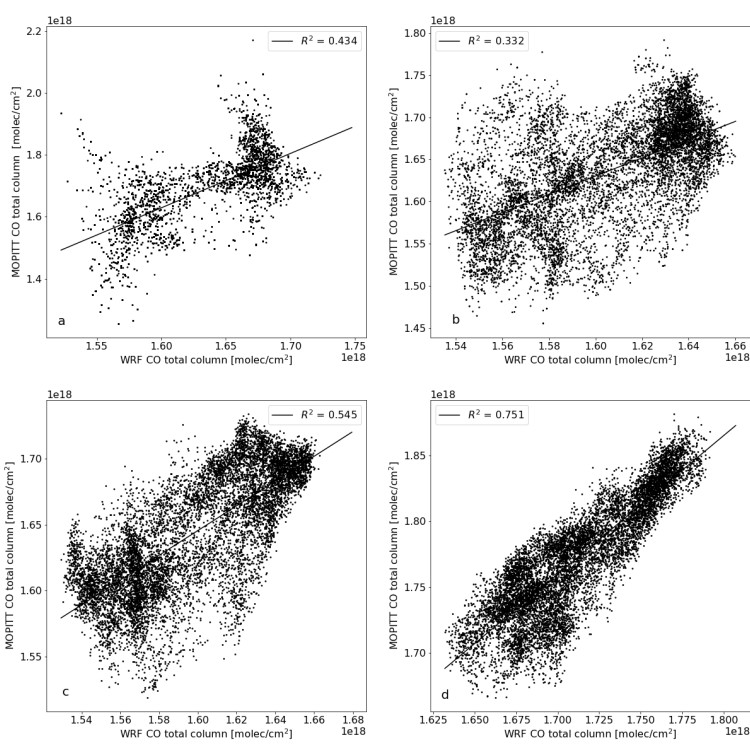

**Figure A6.** Comparison between MOPITT V6 and WRF for different temporal sampling times. WRF results are sampled according to the coordinates of single MOPITT retrievals and both are averaged on a 2x2km$^2$ grid, (a) for a 10 day period (1-10 July 2006), (b) for a 1 month period (July 2006), (c) for a 4 month period: June-September 2006 and (d) for the whole year 2006.





**Table A1.** MOPITT V6 multispectral Downwind−upwind differences ($V_d - V_u$) in total column CO over large cities and the relative difference (RD) between 2000-2003 and 2004-2008, comparing results from this study and Pommier et al. (2013).

| Megacity (Coordinates) | $V_d - V_u$: Our study, (Pommier et al.) 2000-2003 [$10^{17}$ molec/cm$^2$] | $V_d - V_u$: Our study, (Pommier et al.) 2004-2008 [$10^{17}$ molec/cm$^2$] | RD: Our study, (Pommier et al.) [%] |
|---|---|---|---|
| Moscow (55.75°N,37.62°E) | 3.190.04 (2.80.03) | 2.080.04(2.30.06) | −34.933.1 (−18.53.7) |
| Paris (48.86°N,2.36°E) | 1.290.02 (1.30.05) | 0.940.03 (1.00.03) | −27.34.4 (−22.26.9) |
| Mexico (19.43°N,99.13°W) | 6.9770.05 (7.00.09) | 5.340.05 (4.20.06) | −23.381.6 (−39.92.6) |
| Tehran (35.70°N,51.42°E) | 4.050.06 (4.40.02) | 3.0440.02 (2.50.06) | −24.812.0 (−42.92.8) |
| Baghdad (33.33°N,44.38°E) | 2.240.03 (2.20.01) | 1.370.02 (1.20.03) | −39.02.8 (−46.52.9) |
| Los Angeles (34.05°N,118.2°W) | 5.750.06 (6.10.11) | 3.320.117 (4.90.07) | −36.63.6 (−19.63.4) |
| Sao Paulo (23.54°S,46.64°W) | 1.700.02 (1.50.04) | 2.380.08 (1.10.03) | +40.04.4 (−26.95.4) |
| Delhi (28.61°N,77.21°E) | 1.090.02 (0.90.02) | 1.110.02 (1.10.04) | +2.245.6 (+22.45.8) |
| Madrid* (40.41°N,3.71°W) | 0.970.03 (–) | 0.640.02 (–) | −33.05.7 (–) |

*Madrid was not included in the study of Pommier et al. (2013)



**Table A2.** MOPITT V5 multispectral Downwind−upwind differences ($V_d−V_u$) in total column CO over large cities and the relative difference (RD) between 2000-2003 and 2004-2008, comparing results from this study and Pommier et al. (2013).

| Megacity (Coordinates) | $V_d - V_u$: Our study, (Pommier et al.) 2000-2003 [$10^{17}$ molec/cm$^2$] | $V_d - V_u$: Our study, (Pommier et al.) 2004-2008 [$10^{17}$ molec/cm$^2$] | RD: Our study, (Pommier et al.) [%] |
|---|---|---|---|
| Moscow (55.75°N,37.62°E) | 2.41±0.04(2.8 ± 0.03) | 1.74±0.05(2.3 ± 0.06) | −27.9±4.5(-18.5 ± 3.7) |
| Paris (48.86°N,2.36°E) | 1.48±0.06(1.3 ± 0.05) | 0.58±0.03(1.0 ± 0.03) | −60.7±8.5(-22.2 ± 6.9) |
| Mexico (19.43°N,99.13°W) | 7.27±0.06(7.0 ± 0.09) | 5.08±0.04(4.2 ± 0.06) | −30.1±1.6(-39.9 ± 2.6) |
| Tehran (35.70°N,51.42°E) | 5.06±0.05(4.4 ± 0.02) | 3.20±0.03(2.5 ± 0.06) | −21.5±2.6(-42.9 ± 2.8) |
| Baghdad (33.33°N,44.38°E) | 2.31±0.03(2.2 ± 0.01) | 1.23±0.04(1.2 ± 0.03) | −46.7±4.4(-46.5 ± 2.9) |
| Los Angeles (34.05°N,118.2°W) | 4.82±0.07(6.1 ± 0.11) | 3.38±0.07(4.9 ± 0.07) | −29.8±3.7(-19.6 ± 3.4) |
| Sao Paulo (23.54°S,46.64°W) | 1.96±0.03(1.5 ± 0.04) | 1.79±0.05(1.1 ± 0.03) | +5.7±4.9(-26.9 ± 5.4) |
| Delhi (28.61°N,77.21°E) | 1.16±0.02(0.9 ± 0.02) | 1.42±0.04(1.1 ± 0.04) | +22.0±4.3(+22.4 ± 5.8) |
| Madrid* (40.41°N,3.71°W) | 0.79±0.02(−−) | 0.95±0.02(−−) | +20.5±4.6(−−) |

*Madrid was not included in the study of Pommier et al. (2013)





*Competing interests.* The authors declare that they have no conflict of interest.

*Acknowledgements.* We would like to thank Michiel van der Molen en Ingrid Super for their assistance in using the WRF model and the use of their computer infrastructure and SurfSARA for the use of the Cartesius supercomputing system. The MOPITT data were freely obtained

720   from the NASA Langley Research Center Atmospheric Science Data Center. We would also like to thank the Comunidad de Madrid for freely using their air quality network data.