# Peer review of "Quantification of CO emissions from the city of Madrid using MOPITT satellite retrievals and WRF simulations"

_Atmospheric Chemistry and Physics, 2017_

## Referee Comment (RC1) · Anonymous Referee #1 · 25 Jul 2017

This manuscript presents a quantification of CO emissions over Madrid, based on MO-PITT measurements and WRF simulations. In my opinion, this paper represents an interesting work and it is a good complementary work of the study done by Pommier et al. (2013). I was very interested to read this paper especially by thinking that it is a good idea to use a model to optimize the estimation of the emissions. This part was lacking in the work previously done by Pommier et al. (2013). This paper fits perfectly for this journal. Nevertheless, the manuscript is not well structured, making sometimes difficult to read. Thus I recommend publication in ACP after the comments below are addressed.

[Figure]

Structure: Section 2.3.5: I do not understand why you present Fig A6 before A3, A4. . .
The problem of organization is also shown with the caption in Fig A1. At this stage I
do not know what the correction factor is. This factor is only mentioned from page 9.
Moreover, in Fig. A1 the caption and the colors of the curves do not match. There is
no dotted line.

It is odd to finish the paper by the sensitivity tests. These tests should be done before
to analyze the results of the WRF optimization method.

Page 10: Table A2 is described before Tab. A1.

Difference with P13: The authors concluded – quoting the text: "the emission proxies
in P13 are too optimistic". In the same time, they wrote that the RD can change up
to 25% due to the mis-location of the city center. With a quick check with the work
done by Pommier et al. (2013), we can see that the locations of the city used in this
work do not match perfectly with the coordinates used in P13. For example, Sao Paulo
is 23.54S, 46.64W in your work and 23.53S, 46.62W in P13. This represents only a
difference of 2 km but it seems even a difference of 0.7 km has an impact on the RD.
It is interesting to see that P13 did not take into account this problem of location. It
is probably a missing source of error in their study. Thus I agree with the authors the
uncertainties in P13 are probably underestimated.

Another remark about the differences between both studies: the differences may be
explained by 3 parameters: - The resolution of the wind are not similar (0.75 in P13
vs 1deg in this work) - The PBLH (750 hPa in P13 vs 700 hPa in this study) - The
filter used for the MOPITT data (cloud fraction = 0 and cloud index = 2 in P13 vs cloud
diagnostic = 1, 2 in this study). How do use the pixels where there is a conflict between
sea surface and land? P13 filtered out these data. The discrepancy between both
studies may decrease if similar criteria are used.

Other major comments:

Introduction: Is there any publications about the CO trend/pollution over Madrid? It will be informative to have a comparison of your results with previous studies.

Page 3, line 1: Pommier et al. 2013 did not quantify emissions. Estimate the change in the emissions is more appropriate. Clerbaux et al. 2008 did not calculate the emissions but they detected urban CO plumes. Thus delete this reference for this sentence. Then you can write, "Clerbaux et al. (2008) and Pommier et al. (2013) already demonstrated that . . ."

Page 7, line 11: does it means you exclude the first days of your run? What is the period of your simulations? You should introduce this information before Section 2.3.5.

Page 7, line 5: the climatological data, is it for the column or the profile? I guess it is the profile. Please provide the information.

Page 8, lines 26-27. There is a repetition of this information: "background simulation without emissions". Please rephrase.

Page 9, lines 24-27: It is not clear. Please rephrase.

Page 10: I am not sure to fully understand your discrepancy ($0.5 \times 10^{17}$ molecules/cm2). If I average the absolute difference between Vd-Vu from your study and Vd-Vu from P13 in 2000-2003, I find $0.45 \times 10^{17}$ molecules/cm2. Is it the calculation done? Please clarify this point. Same question with 0.009 and 1.04 as I do not find these values in Tab. A1.

Page 10, line 20: -20%: where does the number come from?

Tab. A1 There is an error with the numbers. I think it is for example Moscow: $3.19 \pm 0.04$ The "$\pm$" is missing everywhere.

Page 11, Sect 3.2.1. Did you test your results by excluding 2000 and 2001 since there is a lack of data (i.e. Jan-Feb 2000 and June-July 2001)?

Line 14: What does it mean? "For example, a year with below average cloud cover. . ."

Page 13, line 4: "AK is scaled". It is confusing. You should specify that you are scaling an artificial AK for your test. During my first reading, I understood you wanted to artificially change the MOPITT AKs.

Tab1 why there is only a few values underlined? Do you want to highlight something?

Page 17, line 24: 32% and in Tab A1 it is 33%

Line 26 "with the increase estimated using the WRF optimization method" and in line 21, it is written -8%. Please clarify.

Fig 7. C and F are similar. Please check if the maps are correct.

Page 30. What is this paragraph below figure 7?

Fig9. Write in the caption the difference between both panels.

Figs. A1 and Fig.A2: Add statistical values for the comparison: correlation coefficient, NMB, etc.

Fig A1. Please improve the resolution of this figure.

Page 33 and Fig. A6. Why there are less data in Figs. A6a and A6b. I think it is due to the lack of observations related to the period of the measurements. So please write the number of observations available for the comparison for each plot. What these 10000 points refer to? It is confusing. The differences between MOPITT and WRF could also be related to the difference of the initial horizontal resolution (22km $\times$ 22km at nadir for MOPITT and $0.125° \times 0.0625°$ in the model).

Last line Appendix B. It is the same sentence in Sect 2.3.5. Do not need to repeat twice.

Minor comments: Page 2, line3: quality, spatial resolution

Line 6: (e.g., Beirle et al.,2011; Liu et al.,2016). Line 15: (e.g., Holloway et al., 2007; Khalil and Rasmussen,1990) Line 34: (e.g., Hooghiemstra et al.,2012a; Leeuwen van

et al., 2013; Hooghiemstra et al. 2012b; Girach and Nair, 2014; Yin et al., 2015; Jiang et al., 2017) Same thing for page 4, line 8 – page 8, line 19.

Page 2, line 10: at ground level at high concentration

Line 16: CO is also highly dependent on seasonal variation.

Page 4, line 2: (Deeter et al., 2013; 2014) Line 3: vegetation - Deeter et al., 2009) Line 8: Deeter et al. (2014; 2016).

Page 7, line 16: we used emissions from the EdgarV4.2

Page 8, line 21: "coarser spatial resolutions": Please provide these resolutions.

Page 10, line 34: weighting

Page 11, line 20: need to correct the numbers: 10ˆ16 10ˆ17

Page 12, line 24: (from surface to 800 hPa).

Page 12, line 25 & Fig. A4: AK area. Do you mean AK vector?

Page 15: problem in inversion studies (Jacob et al., 2016).

Page 20 line 8: Do not begin the sentence with "Or,"

Fig1. Please add the location of Madrid on the map.

Figs. 2 & 5. It is very nice and interesting.

figA6. Add labels (a), (b), (c) and (d) on the scatterplots.

Tab. A1 & A2. Write: ". . . from this study and Pommier et al. (2013). The values from Pommier et al. (2013) are provided in parenthesis".
* * *

---

## Referee Comment (RC2) · Anonymous Referee #2 · 10 Aug 2017

The paper presents a new method for estimating mega-city emissions from satellite data in combination with a chemical transport model. It goes beyond the method presented by Pommier et al. (2013) where satellite data only were used to estimate emission trends. In general the paper is well written, and I recommend publication after the following concerns have been addressed.

General Comments:

The relatively large differences between the results presented in the manuscript using MOPITT V5 data and those in Pommier et al. (2013) should be discussed more systematically. Are those differences only due to differences in the wind direction (surface

[Figure]

– 700 mbar averaged winds at 0.75 deg resolution vs. surface – 750 mbar averaged winds at 1 deg. resolution) as mentioned in P10 line 10? It would help to show the differences in winds to those in Pommier et al. (2013); are those larger for LA where the largest discrepancy in downwind minus upwind total column CO is found? In this context also complex topography or coastal effects should play a role, causing winds extracted from analysis files at different resolution to differ more, or even making the choice of an upwind and downwind region within the complex flow invalid. As stated later also a slight change in the rotation point, e.g. related to the imperfect geolocation bias correction applied to the V5 data, causes differences; however the rotation points used in the estimate using V5 data should be identical to Pommier et al. (2013) as the same geolocation bias correction was applied to the data.

The role of the background scaling factor should be made more clear, e.g. by explicitly writing the dependence of the modelled column averages (X_mod[i]) on f_backg and f_emiss, as the model is fully linear this should be straight forward. In this context (i.e. in section 2.3.6) also the sensitivity experiments should be introduced, where changes in "WRF's background emissions" are applied as described in section 3.5.

Appendix: The text for each appendix should include all references to figures and tables included within each appendix. The way the figures are referred to only from within the main text of the manuscript seems to suggest that the figures would be better included in the manuscript itself rather than the appendix.

Specific comments

Pg 8 Ln 16: add a period at the end of the sentence

Pg 8 Ln 22: Please add the notion that the r-square value measures the explained spatial variance of the annually averaged column mole fractions (if I got this right).

Pg 8 Ln 32: "both backgrounds" please explicitly state what those two different background fields are.

Pg 9, Ln 13: "to still maximize the available information" this is unclear; why does using column average mixing ratios maximise the information?

Pg 10 Ln 6: table A2 is referred to before table A1

Pg 10 Ln 35: replace "weighing" by "weighting"

Fig. 5: I suggest to separate the two time periods by colour, and the three different rotation points by symbol shape. This would make it easier to read the figure.

Pg 14 after line 20: the line numbering is incorrect, also on the following pages; I will use the indicated line numbers in the following

Pg 15 Table 1: the table needs reformatting, e.g. use shorter descriptions or labelling for the filters applied (column 4) to shorten the table

Pg 17 Ln 39: 20x20 "optimization method" should be mentioned in the methods section under 2.3.6; why does the change from 2x2 km to 20x20 km have such impact, given the MOPITT resolution of 22 km?

P18 Ln 16: "changing WRF's background emissions" what is meant by that? Section 2.3.6 does not give any clue on what "background emissions" could mean.

P18 Ln 25: "replacing the normal background simulation, without emissions, with a background simulation that has emissions in the area outside the optimisation area" this seems to be in conflict with the statement in section 2.3.6 (P8 Ln 28-30) where it is mentioned that emissions outside of the 200x200 km box around Madrid are already used in the standard case.

Pg 22 Ln 7: the Jacob et al. (2016) has been published as a final paper

Pg 33 29: What is specifically meant by the "oversampling method"? Does that include the rotation of the grid according to wind direction? If so, which wind was taken for the rotation of the WRF grid at each time step, WRF winds or ECMWF winds at 1 deg. as for the MOPITT observations? This needs to be clearly stated so that the reader can

follow what has been done.

Pg 33 line 36: "the stability of the model" may be reformulate to "a lack of spatial variability in the model"

Pg 33, last two sentences: those sentences are repeated from page 8 and should be removed

Pg 35, Fig. A1: The observations seem to have a vary coarse resolution, as indicated by jumps with a step width of 0.1 mg/m3 (corresponding to about 90 ppb). As the background during summer months is about 80 ppb, this resolution seems a bit coarse. -> include in discussion, mention at least

Pg 35, caption Fig. A2: Concentrations from only one location are shown, the text should be revised.

Pg 40: values seem to have a second decimal point instead of a +/-

―――――――――――――――――

---

## Author Response (AR1)

*Thank you for your effort and valuable comments on our paper. Our responses are embedded below in blue.*

This manuscript presents a quantification of CO emissions over Madrid, based on MOPITT measurements and WRF simulations. In my opinion, this paper represents an interesting work and it is a good complementary work of the study done by Pommier et al. (2013). I was very interested to read this paper especially by thinking that it is a good idea to use a model to optimize the estimation of the emissions. This part was lacking in the work previously done by Pommier et al. (2013). This paper fits perfectly for this journal. Nevertheless, the manuscript is not well structured, making sometimes difficult to read. Thus I recommend publication in ACP after the comments below are addressed.

Structure: Section 2.3.5: I do not understand why you present Fig A6 before A3, A4...
*We chose to not refer to Fig A6 in section 2.3.5 since there is some information needed to understand this scatterplot correctly. The information is given in Appendix B, so this is where we introduce Fig A6, at the end of the document. We also moved some of the figures from the appendix to the main text, so there are no conflicts in the order of the presentation in our revised version.*

The problem of organization is also shown with the caption in Fig A1. At this stage I do not know what the correction factor is. This factor is only mentioned from page 9. Moreover, in Fig. A1 the caption and the colors of the curves do not match. There is no dotted line. *Thank you for pointing this out. We changed the caption in the revised version and also changed the figure it now includes only the original WRF data without correction factor.*

It is odd to finish the paper by the sensitivity tests. These tests should be done before to analyze the results of the WRF optimization method. *We changed the order of presenting our paper in the revised version. The sensitivity tests are following the description of the WRF optimization method and are described in an extra section: 3.3.2.*

Page 10: Table A2 is described before Tab. A1. *we corrected the sequence.*
Difference with P13: The authors concluded – quoting the text: "the emission proxies in P13 are too optimistic". In the same time, they wrote that the RD can change up to 25% due to the mis-location of the city center. With a quick check with the work done by Pommier et al. (2013), we can see that the locations of the city used in this work do not match perfectly with the coordinates used in P13. For example, Sao Paulo is 23.54S, 46.64W in your work and 23.53S, 46.62W in P13.
*Thank you for catching that mistake. The wrong coordinates were still in the tables. We updated the coordinates in our calculations to match them with Pommier, but forgot to update the coordinates listed in the tables. We updated the coordinates in the revised version.*

This represents only a difference of 2 km but it seems even a difference of 0.7 km has an impact on the RD. It is interesting to see that P13 did not take into account this problem of location. It is probably a missing source of error in their study. Thus I agree with the authors the uncertainties in P13 are probably underestimated. Another remark about the differences between both studies: the differences may be explained by 3 parameters: - The resolution of the wind are not similar (0.75 in P13 vs 1deg in this work) - The PBLH (750 hPa in P13 vs 700 hPa in this study) – The filter used for the MOPITT data (cloud fraction = 0 and cloud index = 2 in P13 vs cloud diagnostic = 1, 2 in this study). How do use the pixels where there is a conflict between sea surface and land? P13 filtered out these data. The discrepancy

between both studies may decrease if similar criteria are used. *Thank you for considering these sources of differences. We have addressed these issues in some more detail in the revised version. We agree that the discrepancy might decrease if the exact same criteria were applied. The point we want to make in this section is that the method is very sensitive to slight differences in the filtered data. We did some extra tests to find out the importance of the PBLH and cloud fraction which we included in a new section: "Other sources of uncertainties". We used all the pixels which were according to the MOPITT filter land data, but we agree there could be a problem at the boundary of sea surface and land. We added the following sentence in the section "other sources of uncertainties":*

We do not filter MOPITT data for retrievals containing water bodies other than rejecting water and mixed retrievals using the standard MOPITT flags. Since MOPITT is not able to measure CO in the near-infrared over areas with low albedo, such as water, this can lead to biases in the emission trend estimates in our method. For Los Angeles and Sao Paulo, which are both close to the coast, our analysis may include some scenes with fractional areas of water, while P13 filtered these out. This might explain part of the difference in RD estimation seen in Fig. 5, especially for Sao Paulo.

Other major comments: Introduction: Is there any publications about the CO trend/pollution over Madrid? It will be informative to have a comparison of your results with previous studies.
*Unfortunately we could not find any study on CO trend or other pollution over Madrid*

Page 3, line 1: Pommier et al. 2013 did not quantify emissions. Estimate the change in the emissions is more appropriate. Clerbaux et al. 2008 did not calculate the emissions but they detected urban CO plumes. Thus delete this reference for this sentence. Then you can write, "Clerbaux et al. (2008) and Pommier et al. (2013) already demonstrated that…"
*We changed the text according to your comments:*
Furthermore, the first attempts have been made to use MOPITT CO retrievals to estimate emission changes over cities (Pommier et al., 2013). Clerbaux et al. (2008) and Pommier et al. (2013) demonstrated that CO pollution plumes over large cities can be distinguished from the background in satellite data.

Page 7, line 11: does it means you exclude the first days of your run? What is the period of your simulations? You should introduce this information before Section 2.3.5.
*No, we did not exclude the first days of our simulation. We did try this but did not find a significant difference in the yearly average values when excluding the first days of the run. We added the following sentence in section 2.3.1:*
Our WRF simulations were covering exactly one year, either 2002 or 2006.

Page 7, line 5: the climatological data, is it for the column or the profile? I guess it is the profile. Please provide the information.
*It is for the profile. We clarified this in the paper by adding "profiles" in the text:*
The CO boundary conditions of the outer domain were based on MOPITT profiles of climatological retrieved data.

Page 8, lines 26-27. There is a repetition of this information: "background simulation without emissions". Please rephrase.

*We changed the text to take out the repetition and we now only describe the standard background simulation in the subsection "From model mixing ratios to emission", the other background simulation was described in the section "Sensitivity tests":*

For each year also a background simulation was performed where the boundary and initial conditions are kept the same as in the simulations with emission but where emissions were switched off. The difference between these simulations represents the contribution of the emissions of Madrid to the simulated CO concentrations.

*We added the following in the paragraph on sensitivity tests:*

Extra background simulations were performed in order to quantify this effect: simulations with emissions outside of the 200x200 $km^2$ box around Madrid, and, as the normal simulation, without emissions in the urban area where the optimizations were performed.

Page 9, lines 24-27: It is not clear. Please rephrase.
*We rephrased the sentences. We hope the text is clear now:*

Four different filtering methods were tested to prevent outliers in the MOPITT data to influence the estimation: 1) Filtering out all MOPITT data that were more than three or 2) four standard deviations from the yearly 200x200 $km^2$ mean MOPITT CO concentration, or filtering out all MOPITT and WRF data at the same time and location that had a larger difference between them than 3) three (which is the default method) or 4) four standard deviations from the mean difference be- tween MOPITT and WRF at the same time and location.

Page 10: I am not sure to fully understand your discrepancy ($0.5*10^{17}$ molecules/$cm^2$). If I average the absolute difference between Vd-Vu from your study and Vd-Vu from P13 in 2000-2003, I find $0.45*10^{17}$ molecules/$cm^2$. Is it the calculation done? Please clarify this point. Same question with 0.009 and 1.04 as I do not find these values in Tab. A1.
*$0.5286*10^{17}$ is the mean difference between Vd-Vu from our study with MOPITT V5 data and P13 for both 2000-2003 and 2004-2008, thus comparing each city for both time periods our study and P13 and then calculating the mean for all cities and both time periods.*
*0.00883 is the minimum difference we found between our results with V5 and V6 data: the difference in Vd-Vu between V5 and V6 for Sao Paulo 2000-2003.*
*1.014 (and not 1.04, typing error) is the maximum difference we found between our study V5 and V6: Tehran 2000-2003.*
*We changed the text slightly to clarify:*

When the results of our approach are compared between using V5 and V6 of the data (compare Table A1 with Table A2), we find absolute discrepancies between $0.009 \times 10^{17}$ and $1.014 \times 10^{17}$ molecules/$cm^2$ with an average discrepancy of $0.3 \times 10^{17}$ molec/$cm^2$ .

Page 10, line 20: -20%: where does the number come from?
*This is a tilde: "~"20%, meant to indicate differences of around 20%. This is a rough estimate of the difference between V5 of our study and P13 in Figure 2. We changed*

*uncertainty to difference in the text to make this clear:*
The RD estimations, however, do agree with an absolute difference of ~20% for most cities, so the method still has some value to make a rough estimation of trends in a simple and fast way.

Tab. A1 There is an error with the numbers. I think it is for example Moscow: 3.19_0.04 The "±" is missing everywhere.
*Something went wrong indeed with copying the table to LaTeX. We included the ± .*

Page 11, Sect 3.2.1. Did you test your results by excluding 2000 and 2001 since there is a lack of data (i.e. Jan-Feb 2000 and June-July 2001)?
*No, we did not test this. As can be seen in Figure 3, left side, the variations in average total columns are indeed largest for 2000 and 2001. On the right side of the figure we show the downwind-upwind differences per year. The variation is very large, but 2000 and 2001 are not distinguishable as different from the other years. The point we want to make here is that temporal and spatial sampling differences between years can make an important difference in downwind-upwind differences. The exclusion of 2000 and 2001 would, in our opinion, not add additional information on this point.*

Line 14: What does it mean? "For example, a year with below average cloud cover…"
*We are not sure that we understand your question correctly here. We mean a year that is less cloudy in the summer than an average year. We hope we made it clear in the text now:*
For example, a year with fewer overcast days in summer than an average year …

Page 13, line 4: "AK is scaled". It is confusing. You should specify that you are scaling an artificial AK for your test. During my first reading, I understood you wanted to artificially change the MOPITT AKs.
*We changed the text now to clarify:*
For Madrid, we tested this by constructing a synthetic dataset of MOPITT retrievals for the years 2000 to 2008, all based on WRF-Chem simulated CO vertical profiles over Madrid for 2002 sampled at MOPITT time and location. For each year, we constructed artificial AKs based on the MOPITT AKs. Every AK is scaled such that the annual mean sensitivity remains at the level of 2002 for each AK layer. This led to a negative difference in RD of −5% compared to the same calculation with original AKs.

Tab1 why there is only a few values underlined? Do you want to highlight something?
*Indeed we wanted to highlight the method we use as standard method. We clarified this in the text:*
The results of these tests are summarized in Fig. 8 and Table 1. The results of the default procedure that are shown as blue triangles in Fig. 9 are underlined in Table 1.

Page 17, line 24: 32% and in Tab A1 it is 33%
*Changed, both should be 33%*
Line 26 "with the increase estimated using the WRF optimization method" and in line 21, it is written -8%. Please clarify.
*We found indeed a decrease using the WRF optimization method with the standard filtering. Averaged over all sensitivity tests, however, we found a positive trend. Both are stated in line 26 - 28. We changed the text about the agreement to make it more consistent:*
However, when we limit this satellite-only analysis to the years 2002 and 2006,

a 5% emission increase is found ($V_d - V_u = 1.01 \times 10^{17}$ in 2002 and $1.07 \times 10^{17}$ in 2006), which is in better agreement with the small increase estimated with the average of all sensitivity tests of the WRF optimization method and the relatively small decrease estimated with the standard WRF optimization method.

Fig 7. C and F are similar. Please check if the maps are correct.
*The maps are correct and slightly different. The correction factors are very small, which leads to very small differences.*

Page 30. What is this paragraph below figure 7?
*This is part of paragraph 3.3 on emission estimation with the WRF optimization method. We moved it to the right place again.*

Fig9. Write in the caption the difference between both panels.
*We added this information to the caption:*
upper panel: emission estimations based on EdgarV4.2 prior only; lower panel: including other prior emissions in the WRF model for optimization (see text). The uncertainty of the Edgar and MACC emission inventory estimates are estimated at 50%-200% (Kuenen et al., 2014)

Figs. A1 and Fig.A2: Add statistical values for the comparison: correlation coefficient, NMB, etc.
*We added the correlation coefficient, mean absolute error and root mean square error.*

Fig A1. Please improve the resolution of this figure. *done*

Page 33 and Fig. A6. Why there are less data in Figs. A6a and A6b. I think it is due to the lack of observations related to the period of the measurements. So please write the number of observations available for the comparison for each plot. What these 10000 points refer to? It is confusing. The differences between MOPITT and WRF could also be related to the difference of the initial horizontal resolution (22km _ 22km at nadir for MOPITT and 0.125_0.0625_ in the model).
*As is described in Appendix B, all subplots contain the same amount of data. There are $100x100$ grid cells of $2x2km^2$ on which the data of MOPITT and WRF is gridded using the oversampling technique. In the shorter periods there are grid cells that contain exactly the same information as the neighboring cells, leading to more overlapping points.*

Last line Appendix B. It is the same sentence in Sect 2.3.5. Do not need to repeat twice.
*We deleted the double information in Appendix B.*

Minor comments:
*Thank you for noting, we changed our text as suggested, except if otherwise stated*
Page 2, line3: quality, spatial resolution *done*
Line 6: (e.g., Beirle et al.,2011; Liu et al.,2016). Line 15: (e.g., Holloway et al., 2007; Khalil and Rasmussen,1990) Line 34: (e.g., Hooghiemstra et al.,2012a; Leeuwen van et al., 2013; Hooghiemstra et al. 2012b; Girach and Nair, 2014; Yin et al., 2015; Jiang et al., 2017) Same thing for page 4, line 8 – page 8, line 19. *done*
Page 2, line 10: at ground level at high concentration *done*
Line 16: CO is also highly dependent on seasonal variation. *This is noted in Line 15.*
Page 4, line 2: (Deeter et al., 2013; 2014) Line 3: vegetation - Deeter et al., 2009) Line 8: Deeter et al. (2014; 2016). *done*

Page 7, line 16: we used emissions from the EdgarV4.2 *done*
Page 8, line 21: "coarser spatial resolutions": Please provide these resolutions. *done: 0.1x0.1 degree*
Page 10, line 34: weighting *done*
Page 11, line 20: need to correct the numbers: 10ˆ16 10ˆ17 *done*
Page 12, line 24: (from surface to 800 hPa). *done*
Page 12, line 25 & Fig. A4: AK area. Do you mean AK vector? *No we did not mean AK vector. We described the AK area, as is done first by Rodgers (2000), after the colon in line 25.*
Page 15: problem in inversion studies (Jacob et al., 2016). *done*
Page 20 line 8: Do not begin the sentence with "Or," *done*
Fig1. Please add the location of Madrid on the map. *done*
Figs. 2 & 5. It is very nice and interesting. *Thank you*
figA6. Add labels (a), (b), (c) and (d) on the scatterplots. *These labels are already included in the lower left corner*
Tab. A1 & A2. Write: "… from this study and Pommier et al. (2013). The values from Pommier et al. (2013) are provided in parenthesis". *done*

Anonymous Referee #2

*Thank you for your effort and valuable comments on our paper. Our responses are embedded below in blue.*

The paper presents a new method for estimating mega-city emissions from satellite data in combination with a chemical transport model. It goes beyond the method presented by Pommier et al. (2013) where satellite data only were used to estimate emission trends. In general the paper is well written, and I recommend publication after the following concerns have been addressed.
General Comments:

The relatively large differences between the results presented in the manuscript using MOPITT V5 data and those in Pommier et al. (2013) should be discussed more systematically. Are those differences only due to differences in the wind direction (surface – 700 mbar averaged winds at 0.75 deg resolution vs. surface – 750 mbar averaged winds at 1 deg. resolution) as mentioned in P10 line 10? It would help to show the differences in winds to those in Pommier et al. (2013); are those larger for LA where the largest discrepancy in downwind minus upwind total column CO is found? In this context also complex topography or coastal effects should play a role, causing winds extracted from analysis files at different resolution to differ more, or even making the choice of an upwind and downwind region within the complex flow invalid.

*Thank you for these remarks. We performed some extra tests to investigate the influence of the differences between our study and Pommier et al. We added a new paragraph to describe other differences between our study and Pommier et al., and the possible influence on the emission trend estimation. We agree that complex topography and coastal effects might also influence the estimation and can be somewhat different between P13 and our study due to resolution differences of the wind data. The point we want to make in this section is that the method is very sensitive to slight differences in the filtered data.*

**Other sources of uncertainties**
Since we used a slightly different pressure level for top of the boundary layer (BL) than P13 to calculate the average wind direction, we tested the sensitivity of the relative difference calculation to the height over which the wind-direction was averaged. For this test we took the average over 12 (low BL), 15 (normal BL) or 18 (high BL) hybrid pressure layers, respectively at an average pressure of 808 hPa, 717 hPa and 613 hPa. The height of the averaging was found quite important in determining the value of the RD. For some cities, the differences were rather small, but for Moscow, Paris, Sao Paulo and Delhi, significant differences were found between the RD values for the calculations using different pressure layers. We found absolute differences of over 20%, and an opposite trend sign for Delhi, where the downwind - upwind difference between the two periods is rather small. Just as was found for the dependence on the location of the rotation point, the downwind-upwind emission estimation values are usually quite close to each other, but the difference between 2000-2003 and 2004-2008 is relatively small compared to the spread in downwind-upwind values of one period, leading to large differences in the RD values, as P13 also described in the supporting information of the paper. From this we conclude that the choice of the height over which the wind direction is averaged is important for the satellite-only technique. Since there is no objective criterion to choose the "best" height for rotating the CO column values, this introduces another systematic source of error that will affect the reliability of the results.

By extending the cloud filtering from data with less than five percent clouds, as we did by filtering on cloud diagnostic 1 or 2, to data with a maximum of zero percent clouds, as in P13, the amount of data is reduced by less than a percent. The emission estimation, however, still changes for some cities. For Paris, the downwind-upwind difference is changing by 27% for the 2004-2008 period. The absolute RD change is around 6% for most cities, although for Delhi a 21% difference was found.

We do not filter MOPITT data for retrievals containing water bodies other than rejecting water and mixed retrievals using the standard MOPITT flags. Since MOPITT is not able to measure CO in the near-infrared over areas with low albedo, such as water, this can lead to biases in the emission trend estimates in our method. For Los Angeles and Sao Paulo, which are both close to the coast, our analysis may include some scenes with fractional areas of water, while P13 filtered these out. This might explain part of the difference in RD estimation seen in Fig. 5, especially for Sao Paulo. As described in the supporting information of P13 also the averaging radius, the size of the grid cells, and the across-wind averaging distance can significantly influence the RD estimation.

As stated later also a slight change in the rotation point, e.g. related to the imperfect geolocation bias correction applied to the V5 data, causes differences; however the rotation points used in the estimate using V5 data should be identical to Pommier et al. (2013) as the same geolocation bias correction was applied to the data.

*There should indeed be no difference between our study and P13 on that point because we used the same location and geolocation bias. Still we think it is important to state that a slight difference might cause a significant RD estimation difference. As we describe in Sec. 3.2.4:* This can be an important reason for the differences in emission trends found between V5 and V6. We note that the geolocation bias correction that was used in P13 and our study was slightly different from the correction done for V6 of the data by the MOPITT team (Deeter, 2012). This is a potential source of error since small location shifts can have a substantial effect on the RD estimation.

The role of the background scaling factor should be made more clear, e.g. by explicitly writing the dependence of the modelled column averages (X_mod[i]) on f_backg and f_emiss, as the model is fully linear this should be straight forward.

*We added the following equation to make the role of the background more clear:*
The $X_{mod}$ is built up from data of the background simulation $X_{backg}$ and the full simulation including emissions $X_{emis}$ according to Eq. 5.

$$X_{mod} = X_{backg} \cdot f_{backg} + (X_{emis} - X_{backg}) \cdot f_{emis} \quad (5)$$

In this context (i.e. in section 2.3.6) also the sensitivity experiments should be introduced, where changes in "WRF's background emissions" are applied as described in section 3.5.

*We added an extra paragraph to introduce the sensitivity experiments directly afterwards:*

In order to determine how sensitive our method is to different spatial averaging, different prior emissions and different filtering methods, we performed some sensitivity tests. We tested the optimization with a 10 times coarser grid, i.e., 20x20 km$^2$ to investigate the sensitivity to the chosen grid size and decrease the importance of patterns in the background and emission. We also used different prior emission

patterns: for 2006 we started the optimization with TNO-MACC-III emissions (Kuenen et al., 2014) for 2002 we did a test optimization starting with emissions of 2006. We also tested the sensitivity to emissions in the direct surroundings of the 200x200 km$^2$. Extra background simulations were performed in order to quantify this: simulations with emissions outside of the 200x200 km$^2$ box around Madrid, and, as the normal simulation, without emissions in the urban area where the optimization was performed.

To analyze the robustness of the method, we repeated the optimization using different data filters and investigated the effect of optimizing the absolute difference instead of the quadratic difference in Eq. 4. Four different filtering methods were tested to prevent outliers in the MOPITT data to influence the estimation: 1) Filtering out all MOPITT data that were more than three or 2) four standard deviations from the yearly 200x200 km$^2$ mean MOPITT CO concentration, or filtering out all MOPITT and WRF data at the same time and location that had a larger difference between them than 3) three (which is the default method) or 4) four standard deviations from the mean difference between MOPITT and WRF at the same time and location.

Appendix: The text for each appendix should include all references to figures and tables included within each appendix. The way the figures are referred to only from within the main text of the manuscript seems to suggest that the figures would be better included in the manuscript itself rather than the appendix. *We agree that some figures are more relevant in the main text, we added Fig. A1-A3 and A5 to the main text.*

Specific comments
*Thank you for noting, we changed our text as suggested, except if otherwise stated:*
Pg 8 Ln 16: add a period at the end of the sentence *done*

Pg 8 Ln 22: Please add the notion that the r-square value measures the explained spatial variance of the annually averaged column mole fractions (if I got this right).
*Yes that is right. We added the following information:*
This $R^2$ value quantifies the fraction of the variance in the MOPITT data that is explained by WRF. We also found a clearly visible enhancement of CO mixing ratio over the city of Madrid for this yearly period.

Pg 8 Ln 32: "both backgrounds" please explicitly state what those two different background fields are.
*We changed the description of the backgrounds to make this more clear:*
For each year also a background simulation was performed where the boundary and initial conditions are kept the same as in the simulations with emission but where emissions were switched off. The difference between these simulations represents the contribution of the emissions of Madrid to the simulated CO concentrations.

*We added the following in the paragraph on sensitivity tests:*
Extra background simulations were performed in order to quantify this: simulations with emissions outside of the 200x200 km$^2$ box around Madrid, and, as the normal simulation, without emissions in the urban area where the optimization was performed.

Pg 9, Ln 13: "to still maximize the available information" this is unclear; why does using column average mixing ratios maximise the information?
*We removed the maximize statement and added the following explanation:*
Using the column data in molec/cm$^2$, as done in P13, is not appropriate here, due to the effects of orography that also influence the match between the model and satellite. Instead, the column average CO mixing ratio was used. Note that we do not use the surface layer CO mixing ratio but the total column since the bias, and bias drift, of the multispectral total column product is much lower than that of one or a few layers near the surface (Deeter et al., 2014).

Pg 10 Ln 6: table A2 is referred to before table A1 *we changed the order of presenting the tables*
Pg 10 Ln 35: replace "weighing" by "weighting" *done*
Fig. 5: I suggest to separate the two time periods by colour, and the three different rotation points by symbol shape. This would make it easier to read the figure. *This is how the figure was already, therefore we did not change it.*
Pg 14 after line 20: the line numbering is incorrect, also on the following pages; I will use the indicated line numbers in the following
Pg 15 Table 1: the table needs reformatting, e.g. use shorter descriptions or labelling for the filters applied (column 4) to shorten the table *we removed the long names in column 4 to make the table smaller and clearer.*
Pg 17 Ln 39: 20x20 "optimization method" should be mentioned in the methods section under 2.3.6; why does the change from 2x2 km to 20x20 km have such impact, given the MOPITT resolution of 22 km?
*The oversampling technique applied to a year of data is giving a quite detailed pattern of CO mixing ratios over Madrid, since most data are sampled at slightly different locations. Optimization on 20x20 km2 uses 100 grid cells instead of the 10000 grid cells of the 2x2 km2 grid. This leads to some grid cells in the low resolution optimization that include both the areas where emission takes place and where no emission takes place, making it better performing for the background but worse for the 'transition zone' between emissions and background which is why it is not surprising that the emission estimations differ.*

P18 Ln 16: "changing WRF's background emissions" what is meant by that? Section 2.3.6 does not give any clue on what "background emissions" could mean.
*We updated the description of the background simulations in section 2.3.6 and added some more explanation in line 16:*
To investigate the contribution of emissions outside the optimization area on the pattern in CO in the optimization area, we performed a sensitivity test (sensitivity 1) replacing the normal background simulation, without any emissions, with a background simulation that has emissions in the area outside the 200x200 km$^2$ optimization area. In the ideal case these "background emissions", i.e., the emissions within the WRF domains around the optimization area, only contribute to the background of the 200x200 km$^2$ area around Madrid without affecting the city pattern. In this case, it is sufficient to optimize the background with only one factor.

P18 Ln 25: "replacing the normal background simulation, without emissions, with a background simulation that has emissions in the area outside the optimisation area" this seems to be in conflict with the statement in section 2.3.6 (P8 Ln 28-30) where it is

mentioned that emissions outside of the 200x200 km box around Madrid are already used in the standard case.

*It was mentioned in this paragraph that "*Most of the results in this paper are therefore based on the simplest setup for the background simulation: the one without any emissions.*", but we realize the description was not so clear. We now changed the description of the background simulation and added a paragraph to explain the sensitivity tests as explained in the answer to your comments on* Pg 8 Ln 32.

Pg 22 Ln 7: the Jacob et al. (2016) has been published as a final paper *we updated the reference*

Pg 33 29: What is specifically meant by the "oversampling method"? Does that include the rotation of the grid according to wind direction? If so, which wind was taken for the rotation of the WRF grid at each time step, WRF winds or ECMWF winds at 1 deg. as for the MOPITT observations? This needs to be clearly stated so that the reader can follow what has been done. *The oversampling method does not include the wind rotation. We explained it better now:*

For each period the oversampling method was applied to grid both WRF and MOPITT data on the 2x2km$^2$ grid; no wind rotation was performed.

Pg 33 line 36: "the stability of the model" may be reformulate to "a lack of spatial variability in the model" *thank you for the suggestion; we reformulated the text this way.*

Pg 33, last two sentences: those sentences are repeated from page 8 and should be removed *done*

Pg 35, Fig. A1: The observations seem to have a vary coarse resolution, as indicated by jumps with a step width of 0.1 mg/m3 (corresponding to about 90 ppb). As the background during summer months is about 80 ppb, this resolution seems a bit coarse. -> include in discussion, mention at least *We added the following sentence in the text:*

It should be noted that the resolution of the observations is 0.1 mg/m$^3$, especially for the background station Villa del Prado, this resolution is close to the absolute value of the measurement (0.1 mg/m3 corresponds to about 90 ppb) and could thus be considered a bit coarse for measuring background concentrations.

Pg 35, caption Fig. A2: Concentrations from only one location are shown, the text should be revised. *We revised the text accordingly, now including only Mostelos.*

Pg 40: values seem to have a second decimal point instead of a +/- *we added the ± sign in the table*

We revised the manuscript according to the above listed comments of the reviewers. Below we list the most important changes that we made in the manuscript:

- We corrected spelling mistakes and corrected some punctuation marks.
- We included figure A1, A2, A3 and A5 in the main text, instead of in the appendix.
- We added an extra paragraph introducing the sensitivity tests in the methods section.
- We changed the order of the methods sections: the paragraph on comparing MOPITT and WRF is in the revised manuscript after the validation of WRF data section.
- We explained the difference between the emission run and background run more clearly.
- We added an equation to explain the dependence of the modelled column averages (X_mod[i]) on f_backg and f_emiss.
- We did some extra tests on the RD dependence of the boundary layer height to average the wind
- We also did some extra tests on the RD dependence on filtering for <5% or 0% clouds. We included both of these tests in the additional section on "other sources of uncertainties".
- We mentioned the uncertainties that Pommier et al. (2013) described in his paper to this section as well.
- We explained the results of the sensitivity tests more clearly now in section 3.3.2: Sensitivity tests. This paragraph is now also found earlier in the manuscript.

[revised manuscript text omitted]
 (red and blue lines) compared to observations (orange and light blue lines) at two locations near Madrid.

[Figure]

**Figure 3.** Hourly WRF surface concentrations (red line) compared to observations (orange line) at the Mostelos measurement station near Madrid for 10 days in October.

tions appear very similar between WRF and the observations (r = 0.75 and r = 0.47 respectively), although WRF overestimates the concentrations at the Villa del Prado station (Fig. 2, upper panel). The variation over the months with higher concentrations in winter is well represented, most peaks seen in the observations are also found in the model and concentration differences between model and observation are generally within 0.1 mg/m$^3$. It should be noted that the resolution of the observations is 0.1 mg/m$^3$, especially for the background station Villa del Prado, this resolution is close to the absolute value of the measurement (0.1 mg/m$^3$ corresponds to about 90 ppb) and could thus be considered a bit coarse for measuring background concentrations. The overestimated CO concentration for the Villa del Prado station is considered reasonable, since with the resolution of 10x10 km$^2$ of WRF, the WRF pixel also includes two small towns in this area, while the station is measuring at a very remote location at the Villa del Prado station. On hourly time scale, WRF also follows the observations quite well (Fig. 3, r = 0.31), stable low concentration patterns are also represented in the model as such and higher concentrations with morning and afternoon peaks are also represented, although WRF is not able to see all peaks and some peaks are under and overestimated (differences of up to 1 mg/m$^3$). Given the limited resolution used in WRF and the difficulty of representing measurement sites in an urban environment, we consider the performance of WRF adequate to make a reasonable comparison with the coarser resolution satellite data. For 2002, only data from the Mostelos station are available. In Fig.2, lower panel, the comparison with these data is shown; the concentrations match also very reasonably for as well the peaks as the yearly patterns (r = 0.73), the concentrations do most of the time overlap within 0.1 mg/m$^3$.

**2.3.4   Comparing MOPITT and WRF**

The information of the MOPITT retrievals is not equally distributed over the 10 vertical levels, as mentioned earlier. For a fair comparison between satellite observations and model simulations, the **AK** matrix and a priori profile for each retrieval has been applied to the corresponding model output, ensuring a consistent vertical weighting of the model compared with the measurements. The MOPITT averaging kernel matrix was applied to the logarithm of model simulated CO concentrations following Eq. 2, using the interpolated vertical model profile of CO from WRF as $x_{true}$, $x_{retr}$ forms then the WRF vertical profile on MOPITT levels with the applied averaging kernel matrix that is used for comparison. In the comparison, average mixing ratios over all vertical MOPITT layers are used. For this method we only used MOPITT V6 data.

**2.3.5   Simulation period**

To reduce the random noise and to increase the signal from relatively small sources, it is required to average MOPITT data over longer time periods as earlier studies already mentioned  (e.g., Clerbaux et al., 2008; Girach and Nair, 2014; Deeter et al., 2014). Averaging times ranged in these studies from 1 month for the second study to 7 years for the first study; it should be noted, however, that these studies used coarser spatial resolutions: 1°x 1°. In our study we chose to average 1 year of data, which resulted in quite good comparison with WRF: R$^2$ = 0.75. This R$^2$ value quantifies the fraction of the variance in the MOPITT data that is explained by WRF. We also found a clearly visible enhancement of CO mixing ratio over the city of Madrid for this yearly period. A description of the more detailed test we did that resulted in the use of a period of a year can be found in Appendix B.

**2.3.6   From model mixing ratios to emissions**

 For comparison with MOPITT, model simulations were done  for the

years 2002 and 2006 with EdgarV4.2 emissions of the corresponding years. For each year also a background simulation  was performed where the boundary and initial conditions are kept the same as in the simulations with emission but where emissions were switched off. The difference between  these simulations represents the contribution of the emissions of Madrid to the simulated CO concentrations.

Since tracer transport in WRF is linear, the CO contribution from Madrid scales linearly with its emission. Because of this, the optimal, i.e., best fit, emission was linked to the inventory emission by a scaling factor ($f_{emis}$) of the simulated urban plume: the difference between CO in the emission and background simulation. To make this method easily applicable to other regions and to limit the required WRF computation time, we implemented only direct anthropogenic CO emissions and assumed a uniform distribution of other sources of CO (e.g., anthropogenic sources of CO in the surroundings, direct natural sources and indirect sources of CO such as the atmospheric oxidation of natural and anthropogenic volatile organic carbon compounds and methane from the city or the surrounding forests). To account for these missing sources in the 200x200 km$^2$ area around Madrid, a background correction factor ($f_{back}$) was introduced that has no spatial pattern but is simply a multiplication factor of the concentrations in the background simulation.

After a WRF simulation, the WRF data were sampled according to the MOPITT retrievals, the **AK** matrix and MOPITT a priori profile were applied, and the mixing ratios were gridded on a 2x2 km$^2$ grid and averaged over the entire column with the oversampling technique of Fioletov et al. (2011), as described in section 2.2 and used in P13.  Using the total column data in molec/cm$^2$, as done in P13, is not appropriate here,  due to the effects of orography that also influence the match between the model and satellite. Instead, the  column average CO mixing ratio was used. Note that we do not use the surface layer CO mixing ratio but the total column since the bias, and bias drift, of the multispectral total column product is much lower than that of one or a few layers near the surface (Deeter et al., 2014).

To estimate CO emissions, we used a simple optimization scheme based on Brent's method (Brent, 1973; Press et al., 1992). We minimized the difference between MOPITT and WRF average column mixing ratios by varying $f_{backg}$ and $f_{emis}$ iteratively using Brent's method. Brent's method is a root finding algorithm, which we used to find the minimum of the quadratic cost function $J$ (ppb$^2$), defined in Eq. 4:

$$J = \sum_{i=1}^{n}((X_{mod[i]}(f_{backg}, f_{emis}) - X_{sat[i]})^2) \quad (4)$$

In this function, $n$ is the number of grid cells within the 200x200 km$^2$ optimization domain. $X_{mod[i]}$ is the total column average mixing ratio (ppb) in the i$^{th}$ grid cell of the model and $X_{sat[i]}$ the mixing ratio (ppb) in the corresponding MOPITT grid cell. We filtered out data where the difference between MOPITT and WRF was more than three times the standard deviation of their mean difference to prevent outliers from influencing the emission estimation. The $X_{mod}$ is build up from data of the background simulation $X_{backg}$ and the full simulation including emissions $X_{emis}$ according to Eq. 5.

$$X_{mod} = X_{backg} \cdot f_{backg} + (X_{emis} - X_{backg}) \cdot f_{emis} \quad (5)$$

**2.3.7 Sensitivity tests**

In order to determine how sensitive our method is to different spatial averaging, different prior emissions and different filtering methods, we performed some sensitivity tests. We tested the optimization with a 10 times coarser grid, i.e., 20x20 km$^2$, to investigate the sensitivity to the chosen grid size and decrease the importance of patterns in the background and emission. We also used different prior emission patterns: for 2006 we started the optimization with TNO-MACC-III emissions (Kuenen et al., 2014), for 2002 we did a test optimization starting with emissions of 2006. We also tested the sensitivity to emissions in the direct surroundings of the 200x200 km$^2$. Extra background simulations were performed in order to quantify this: simulations with emissions outside of the 200x200 km$^2$ box around Madrid, and, as the normal simulation, without emissions in the urban area where the optimization was performed.

To analyse the robustness of the method, we repeated the  optimization using different data filters  and investigated the effect of  optimizing the absolute difference instead of the quadratic difference in Eq. 4. Four different filtering methods were tested to prevent outliers in the MOPITT data to influence the estimation: 1) Filtering  out all MOPITT data that were more than three or 2) four standard deviations from the yearly 200x200 km$^2$ mean MOPITT CO concentration,

[Figure]

**Figure 4.** Total column CO concentration downwind minus upwind of selected cities (see methods-section), comparing our study using MOPITT version 5 (squares) and the study of Pommier et al. (2013, triangles). Error bars represent uncertainties calculated according to P13.

more or filtering out all MOPITT and WRF data at the same time and location that had a larger difference between them than 3) three (which is the default method) or 4) four standard deviations from the mean difference between WRF and MOPITT . The default procedure was to minimize quadratic differences and filter out differences of more than three times the standard deviation between WRF and MOPITT and WRF at the same time and location.

**3 Results and discussion**

**3.1 Emission trend estimation and uncertainty based on satellite data only**

[revised manuscript text omitted]

**3.2 Limitations of the satellite-only approach: possible sources of errors and sources of uncertainties**

~~When using only satellite data to estimate emission trends, it is important to consider how satellite data are obtained: the maximum a posteriori retrieval is based on a set of measured radiances, a radiative transfer model, and a model-derived a priori profile. The averaging kernel represents the weighing of the measured signal and the a priori information in the retrieved CO profile (see section 2.1).these termsas well as the importance of the exact location of the wind-turning centre~~ which possibly give errors in the emission trend estimation. We will also look at the influence of choices to filter and rotate the data that lead to uncertainties in the trend estimation. 
[revised manuscript text omitted]
  2002 sampled at MOPITT time and location. For each year,  we constructed artificial **AK**s based on the MOPITT **AK**s. Every **AK** is scaled such that the annual mean sensitivity remains at the level of 2002 for each **AK** layer. This led to a negative difference in RD of $-5\%$ compared to the same calculation with original **AK**s. From this result, we conclude that the stability of the **AK** is influencing the emission trend estimation using the satellite-only method, which introduces an uncertainty when using satellite data from MOPITT and potentially also other instruments. It should be noted, however, that the averaging kernel is quite specific for each retrieval and replacing it by a corrected **AK**, as done here, is justified as a sensitivity test but is not considered a solution to the problem, as indicated by the data description paper published in Deeter (2002).

**3.2.4    The rotation point selection**

In the satellite-only approach, a wind rotation technique is applied to calculate upwind $-$ downwind differences. This technique selects a single point in the  center of the city as rotation point. However, we found that the estimated upwind $-$ downwind differences are sensitive to the location of this rotation point, which is problematic since it is hard to tell what the exact  center of a city is. Moving this rotation point for example from the  center defined by Wikipedia to the  center point defined by Google Maps (GM), which differs 0.7-3.9 km for our selected cities - both locations could be equally well defined as  center - gives downwind$-$upwind differences varying by $0.03 \times 10^{17}$-$0.3 \times 10^{17}$ molec/cm$^2$, corresponding to RDs varying by 8%-25% (Fig.8). As a solution for this problem, we using the weighted emission  center of the city instead of the general  center would be a fairer way to use this method. We tested this for the city of Madrid for the weighted  center point in the TNO-MACC emission inventory and weighted  center point of the EdgarV4.2 emission inventory. We found a positive RD of +3% for the Edgar  center and a negative RD of $-4\%$ for the MACC center, which was located 8 km more southwards. These estimations are probably better estimations of the real trend, since it uses the  center of the emissions instead of the  center of the buildings, but it also shows that this problem is difficult to solve, since the exact  center of emissions is also not known.

The satellite-only method is thus highly sensitive to the selected location of the rotation point, which introduces a large uncertainty in the estimated emission trends. This outcome

[Figure]

**Figure 8.** Upwind $-$ Downwind difference (left axis, orange, green) and Relative Difference calculation (right axis, blue points) for Madrid, Bagdad, Delhi and Moscow using different rotation points within the city center. GM: GoogleMaps location of the center, GM shifted: 5 km shift of this point to another center location, Wiki: Wikipedia location of the center. Wikipedia center points are off by 3.9, 3.1, 2.1 and 0.7 km from the GM center points for Madrid, Bagdad, Delhi and Moscow respectively.

is particularly relevant for the use of MOPITT data, because of a location bias in MOPITT version 5, which has been corrected in version 6. This can be an important reason for the differences in emission trends found between V5 and V6.  We note that the geolocation bias correction that was used in P13 and our study was slightly different from the correction done for V6 of the data by the MOPITT team (Deeter, 2012).  This is a potential source of error since small location shifts can have a substantial effect on the RD estimation.

**3.2.5   Other sources of uncertainties**

Since we used a slightly different pressure level for top of the boundary layer (BL) than P13 to calculate the average wind direction, we tested the sensitivity of the relative difference calculation to the height over which the wind-direction was averaged. For this test we took the average over 12 (low BL), 15 (normal BL) or 18 (high BL) hybrid pressure layers, respectively at an average pressure of 808 hPa, 717 hPa and 613 hPa. The height of the averaging was found quite important in determining the value of the RD. For some cities, the differences were rather small, but for Moscow, Paris, Sao Paulo and Delhi, significant differences were found between the RD values for the calculations using different pressure layers. We found absolute differences of over 20%, and an opposite trend sign for Delhi, where the downwind - upwind difference between the two periods is rather small. Just as was found for the dependence on the location of the rotation point, the downwind-upwind emission estimation values are usually quite close to each other, but the difference between 2000-2003 and 2004-2008 is relatively small compared to the

spread in downwind-upwind values of one period, leading to large differences in the RD values, as P13 also described in the supporting information of the paper. From this we conclude that the choice of the height over which the wind direction is averaged is important for the satellite-only technique. Since there is no objective criterion to choose the "best" height for rotating the CO column values, this introduces another systematic source of error that will affect the reliability of the results. By extending the cloud filtering from data with less than five percent clouds, as we did by filtering on cloud diagnostic 1 or 2, to data with a maximum of zero percent clouds, as in P13, the amount of data is reduced by less than a percent. The emission estimation, however, still changes for some cities. For Paris, the downwind-upwind difference is changing by 27% for the 2004-2008 period. The absolute RD change is around 6% for most cities, although for Delhi a 21% difference was found. We do not filter MOPITT data for retrievals containing water bodies other than rejecting water and mixed retrievals using the standard MOPITT flags. Since MOPITT is not able to measure CO in the near-infrared over areas with low albedo, such as water, this can lead to biases in the emission trend estimates in our method. For Los Angeles and Sao Paulo, which are both close to the coast, our analysis may include some scenes with fractional areas of water, while P13 filtered these out. This might explain part of the difference in RD estimation seen in Fig. 5, especially for Sao Paulo. As described in the supporting information of P13 also the averaging radius, the size of the grid cells, and the across-wind averaging distance can significantly influence the RD estimation.

[revised manuscript text omitted]
 200x200 km$^2$ optimization area. In the ideal case these "background emissions", i.e., the emissions within the WRF domains around the optimization area, only contribute to the background of the 200x200 km2 area around Madrid without affecting the city pattern. In this case, it is sufficient to optimize the background with only one factor. If the emissions do contribute to the pattern, we expect the results to have lower cost function values in the optimum. The impact on the optimized emission of Madrid was, however, well within the estimation uncertainty, as can be seen in Fig. 12 from the difference between solid and dotted lines.

[revised manuscript text omitted]

*Madrid was not included in the study of Pommier et al. (2013)